# Opposite physiological and pathological mTORC1-mediated roles of the CB1 receptor in regulating renal tubular function

Liad Hinden [1], Majdoleen Ahmad[1], Sharleen Hamad[1], Alina Nemirovski[1], Gergő Szanda[2], Sandra Glasmacher[3], Aviram Kogot-Levin[4], Rinat Abramovitch [5,6], Bernard Thorens [7], Jürg Gertsch [3], Gil Leibowitz [4] & Joseph Tam [1✉]

Activation of the cannabinoid-1 receptor ($CB_1R$) and the mammalian target of rapamycin complex 1 (mTORC1) in the renal proximal tubular cells (RPTCs) contributes to the development of diabetic kidney disease (DKD). However, the $CB_1R$/mTORC1 signaling axis in the kidney has not been described yet. We show here that hyperglycemia-induced endocannabinoid/$CB_1R$ stimulation increased mTORC1 activity, enhancing the transcription of the facilitative glucose transporter 2 (GLUT2) and leading to the development of DKD in mice; this effect was ameliorated by specific RPTCs ablation of GLUT2. Conversely, $CB_1R$ maintained the normal activity of mTORC1 by preventing the cellular excess of amino acids during normoglycemia. Our findings highlight a novel molecular mechanism by which the activation of mTORC1 in RPTCs is tightly controlled by $CB_1R$, either by enhancing the reabsorption of glucose and inducing kidney dysfunction in diabetes or by preventing amino acid uptake and maintaining normal kidney function in healthy conditions.

[1] Obesity and Metabolism Laboratory, The Institute for Drug Research, School of Pharmacy, Faculty of Medicine, The Hebrew University of Jerusalem, Jerusalem, Israel. [2] Department of Physiology, Semmelweis University, Budapest, Hungary. [3] Institute of Biochemistry and Molecular Medicine, University of Bern, Bern, Switzerland. [4] Diabetes Unit and Endocrine Service, Hadassah-Hebrew University Medical Center, Jerusalem, Israel. [5] The Wohl Institute for Translational Medicine, Hadassah-Hebrew University Medical Center, Jerusalem, Israel. [6] The Goldyne Savad Institute of Gene Therapy, Hadassah-Hebrew University Medical Center, Jerusalem, Israel. [7] Center for Integrative Genomics, University of Lausanne, Lausanne, Switzerland. ✉email: yossi.tam@mail.huji.ac.il

Diabetes is a chronic disease that is now reaching epidemic proportions[1], and has been described as a catalyst for a number of conditions, most notably cardiovascular disease, retinopathy, and diabetic kidney disease (DKD). The latter affects approximately 30% of patients with diabetes, and it is strongly associated with morbidity and mortality[2]. DKD is manifested by glomerular hypertrophy, transient hyperfiltration, albuminuria, kidney fibrosis, and ultimately a progressive decline in the glomerular filtration rate[3]. Recently, a paradigm shift in our understanding of kidney dysfunction in diabetic patients has emerged, indicating that tubulopathy precedes glomerular alterations[4,5]. The renal proximal tubular cells (RPTCs) are uniquely susceptible to a variety of metabolic and hemodynamic factors associated with diabetes, especially to hyperglycemia. In fact, glucose entry into RPTCs is insulin-independent, making these cells particularly sensitive to the deleterious effects of chronic hyperglycemia in diabetic patients, that in turn, may lead to enhanced $O_2$ consumption and increased hypoxic tubular damage[5]. During normoglycemia, however, the RPTCs account for 25% of whole-body gluconeogenesis, and play a key role in glucose[6] and amino acid[7] reabsorption, thus maintaining adequate blood glucose levels and metabolic homeostasis.

Both the cannabinoid-1 receptor ($CB_1R$) and the mammalian target of rapamycin complex 1 (mTORC1) are considered as mediators of the "Thrifty Phenotype", evolutionarily designed for consuming and storing energy when an excess of nutrients is available[8–10]. This effect is of great importance when mammalians prepare themselves for a "time of need". However, this advantage becomes a double-edged sword for humans in our modern world of satiation and contentment, where overnutrition contributes significantly to the increased prevalence of diabetes and obesity. Both $CB_1R$ and mTORC1 were shown to be involved in kidney pathologies under hyperglycemic conditions[11–16]. For instance, $CB_1R$ has been shown to be upregulated in podocytes, mesangial cells, and RPTCs in diabetes, and its pharmacological inhibition by $CB_1R$ antagonists ameliorates diabetes-induced kidney dysfunction, inflammation, and fibrosis[17–25]. Hyperglycemia also activates mTORC1 in these cells and its inhibition ameliorates oxidative stress, endoplasmic reticulum stress, epithelial-to-mesenchymal transition, inflammation, and fibrosis[26–34]. Stimulation of mTORC1 in response to nutrient flux contributes to tubule-interstitial fibrosis[16,31,35–37] and apoptosis[38]; therefore, mTORC1 has been suggested as a potential therapeutic target for DKD[39]. In fact, a positive association between $CB_1R$ and mTORC1 activities has been previously described in the central nervous system[40–44]. In contrast, peripheral $CB_1R$ antagonism has been shown to decrease glucose-stimulated insulin secretion (GSIS) and gastric ghrelin secretion via activation of mTORC1[45,46]. Interestingly, activation of mTORC1 has been shown to stimulate GSIS under diabetic conditions by elevating the protein levels of the facilitative glucose transporter GLUT2[47].

A growing body of evidence indicates that hyperglycemia induces changes in glucose transport via GLUT2 and may negatively affect kidney function and the associated tubulo-interstitial changes seen in DKD[48,49]. In contrast to the reduced expression of GLUT2 in pancreatic β cells in different models of diabetes[50–52], its levels in RPTCs rise in diabetic patients[53] as well as in murine models of diabetes and obesity[54–56]. Plasma or luminal glucose concentrations are known to regulate its expression and/or translocation[57], accounting for the deleterious effects of hyperglycemia on the proximal tubule. Recently, we have demonstrated that diabetes-induced upregulation in kidney GLUT2 expression and dynamic translocation is mitigated by peripheral pharmacological blockade or genetic deletion of $CB_1R$ in RPTCs, and that it reduces glucose reabsorption and prevents the development of DKD[25]. However, there are several fundamental open questions

that should be addressed: What are the cellular mechanisms by which hyperglycemia activates $CB_1R$? How do $CB_1Rs$ specifically regulate the transcriptional levels of GLUT2? And what is the explicit role of GLUT2 in RPTCs in the development of DKD? Addressing these important questions will eventually advance our understanding of the interaction of $CB_1R$ and GLUT2 in the development of DKD. Based on the key roles of $CB_1R$ and mTORC1 in whole-body energy utilization, and their similar involvement in the pathogenesis of DKD, we aimed to determine whether these two pathways interact in the proximal tubules and affect kidney homeostasis in health and disease. Here, we show a profound inactivation of mTORC1 signaling and, consequently, downregulation of GLUT2 in diabetic mice lacking $CB_1R$ specifically in RPTCs; these molecular events were associated with preserved kidney function in the diabetic mice. Furthermore, we show that hyperglycemia specifically enhances the endocannabinoid (eCB)/$CB_1R$ 'tone' in RPTCs, promoting a cascade of molecular events that activate mTORC1, which in turn, triggers transcriptional changes in the GLUT2 promoter, ultimately enhancing its expression. On the other hand, non-diabetic mice lacking $CB_1R$ in RPTCs display enhanced mTORC1 activation as a result of elevated amino acid transport, causing morphological abnormalities in the kidney and renal dysfunction. These observations reveal the critical role of $CB_1R$ in maintaining normal mTORC1 activity in RPTCs both under pathological and physiological conditions. Finally, we demonstrate that a specific reduction in GLUT2 expression in RPTCs is sufficient to protect diabetic mice from developing DKD, emphasizing this transporter's role in regulating renal glucose homeostasis and the pathogenesis of DKD. Overall, our data describe for the first time an opposing $CB_1R$/mTORC1 signaling axis in healthy and diseased kidney.

## Results

**$CB_1R$ in the RPTCs plays a key role in DKD development.** Overactivation of the kidney eCB/$CB_1R$ system contributes to the development of DKD, and its blockade by $CB_1R$ antagonists ameliorates kidney dysfunction in diabetic mice[17–25]. To specifically decipher the importance of $CB_1R$ in RPTCs in DKD, we crossed RPTC-$CB_1R^{-/-}$ mice[21] with insulin-deficient diabetic $Akita^{Ins2+/C96Y}$ mice, thereby generating diabetic mice lacking $CB_1R$ in RPTCs (Supplementary Fig. 1a–e). The metabolic phenotype of Akita-RPTC-$CB_1R^{-/-}$ mice and of Akita-RPTC-$CB_1R^{+/+}$ controls did not differ and they exhibited similar reduction of body weight, hyperglycemia along with hypoinsulinemia without affecting pancreatic weight (Supplementary Fig. 1f–k). Nevertheless, the Akita-RPTC-$CB_1R^{-/-}$ mice were protected from the deleterious effects of hyperglycemia on their kidneys, exhibiting reduced kidney-to-body weight, urine exertion-to-water consumption, and albumin-to-creatinine (ACR) ratios, reduced proteinuria, albuminuria, and urinary excretion of kidney injury marker 1 (KIM1), as well as improved creatinine clearance (CCr; Fig. 1a–g). The amelioration of kidney function in the Akita-RPTC-$CB_1R^{-/-}$ mice compared to Akita was associated with improved kidney morphological appearance (Fig. 1h–k), along with decreased expression of markers of kidney injury, inflammation, and fibrosis (Fig. 1l–n). Taken together, these findings highlight that RPTC-$CB_1Rs$ play an important role in the pathophysiology of DKD.

**Both GLUT2 and mTORC1 are regulated by proximal tubule $CB_1R$.** Molecularly, the absence of RPTC $CB_1R$ in Akita diabetic mice resulted in the inhibition of kidney mTORC1 activity, manifested by the remarkable reduction in ribosomal protein S6 phosphorylation (pS6; Fig. 2a, d) and associated well with the reduced expression of its upstream regulator phosphorylated active AKT

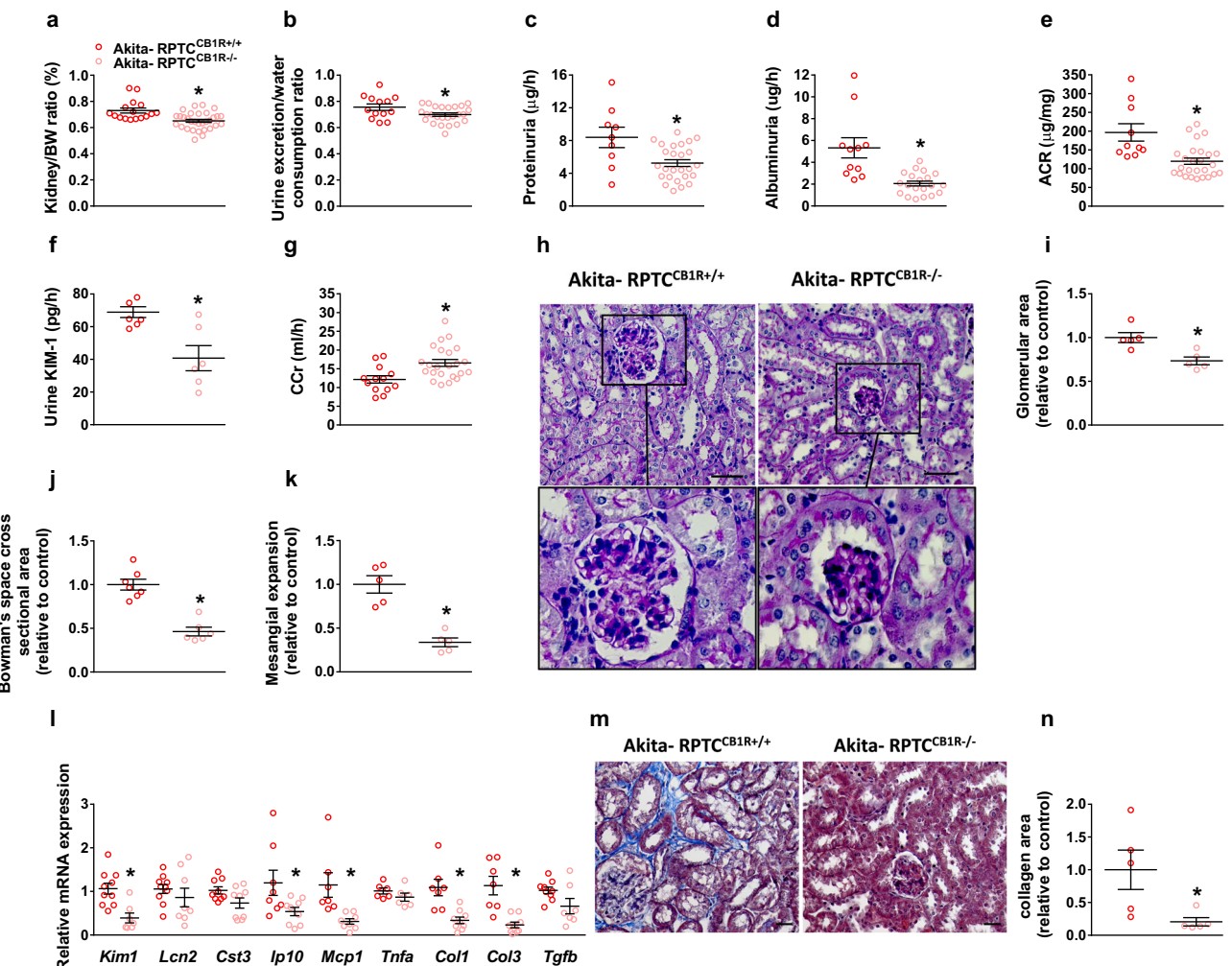

**Fig. 1 Specific CB₁R deletion in RPTCs protects Akita mice from developing DKD. a** Kidney-to-body weight ratio of 16-week-old Akita diabetic mice. $n = 16$ mice for Akita-RPTC$^{CB1R+/+}$, $n = 31$ mice for Akita-RPTC$^{CB1R−/−}$ (*$P = 0.0005$). **b** Urine exertion-to-water consumption ratio of 16-week-old Akita diabetic mice. $n = 10$ mice for Akita-RPTC$^{CB1R+/+}$, $n = 22$ mice for Akita-RPTC$^{CB1R−/−}$ (*$P = 0.0274$). **c** Proteinuria in 16-week-old Akita diabetic mice. $n = 9$ mice for Akita-RPTC$^{CB1R+/+}$, $n = 26$ mice for Akita-RPTC$^{CB1R−/−}$ (*$P = 0.0039$). **d** Albuminuria in 16-week-old Akita diabetic mice. $n = 11$ mice for Akita-RPTC$^{CB1R+/+}$, $n = 17$ mice for Akita-RPTC$^{CB1R−/−}$ (*$P = 0.0274$). **e** Urine albumin-to-creatinine ratio (ACR) in 16-week-old Akita diabetic mice. $n = 10$ mice for Akita-RPTC$^{CB1R+/+}$, $n = 26$ mice for Akita-RPTC$^{CB1R−/−}$ (*$P = 0.0004$). **f** Urine KIM-1 levels in 16-week-old Akita diabetic mice. $n = 7$ mice for Akita-RPTC$^{CB1R+/+}$, $n = 6$ mice for Akita-RPTC$^{CB1R−/−}$ (*$P = 0.0046$). **g** Creatinine clearance (CCr) in 16-week-old Akita diabetic mice. $n = 13$ mice for Akita-RPTC$^{CB1R+/+}$, $n = 23$ mice for Akita-RPTC$^{CB1R−/−}$ (*$P = 0.0040$). **h** Representative PAS staining of the kidney, 40× magnification, scale bar: 50 μm. **i** Glomerular area quantification (at least 10 glomeruli per mouse). $n = 5$ mice for Akita-RPTC$^{CB1R+/+}$, $n = 5$ mice for Akita-RPTC$^{CB1R−/−}$ (*$P = 0.0061$). **j** Bowman's space cross-sectional area quantification (at least 10 glomeruli per mouse). $n = 7$ mice for Akita-RPTC$^{CB1R+/+}$, $n = 6$ mice for Akita-RPTC$^{CB1R−/−}$ (*$P < 0.0001$). **k** Mesangial expansion quantification (at least 10 glomeruli per mouse). $n = 5$ mice for Akita-RPTC$^{CB1R+/+}$, $n = 5$ mice for Akita-RPTC$^{CB1R−/−}$ (*$P = 0.0003$). **l** qPCR analyses of renal injury markers: *Kim1*, *Lcn2*, and *Cst3*, inflammatory markers: *Ip10*, *Mcp1*, and *Tnfa*, and fibrogenic markers: *Col1*, *Col3*, and *Tgfb*. $n = 7$ mice per group (*$P < 0.0363$). **m** Representative Masson's trichrome staining of the kidney, 40× magnification, scale bar: 50 μm. **n** Collagen deposition positive area quantification. $n = 5$ mice per group (*$P = 0.0323$). Data represent the mean ± SEM and were analyzed by Unpaired Two-tailed Student's t-test. Source data are provided as a Source Data file.

(Fig. 2b, e). Importantly, GLUT2 expression was decreased in the Akita-RPTC-CB₁R$^{−/−}$ mice (Fig. 2c, f), concomitant with a significant increase in glycosuria (Fig. 2g). In fact, significant positive correlations between pS6 or pAKT and GLUT2 expression were found (Fig. 2h, i). Conversely, inducing acute CB₁R activation by administering WIN-55,212 to WT mice increased kidney mTORC1 activity and elevated GLUT2 expression levels (Supplementary Fig. 2a–c). These results are in agreement with our previous findings, demonstrating that peripheral pharmacological blockade of CB₁R ameliorates DKD in *Akita* and in streptozotocin (STZ)-induced diabetic mice via downregulating GLUT2 expression and its dynamic translocation[25].

To further study the link between mTORC1 and GLUT2, we utilized the same breeding paradigm to either inactivate or overactivate mTORC1 signaling in RPTCs by generating Akita-RPTC-RPTOR$^{−/−}$ and WT-RPTC-TSC$^{−/−}$ mice[58], respectively. The regulatory-associated protein of mTOR (RPTOR) is essential for mTORC1 activation and its absence in the RPTCs of diabetic mice resulted in reduced pS6 levels (Fig. 2j, k), which were associated with reduced GLUT2 expression (Fig. 2j, l) and resulted in a significant increase in glycosuria (Fig. 2m). These changes were associated with improved kidney function and fibrosis, as recently reported by us in mice with partial inhibition of mTORC1 in RPTCs[58]. On the other hand, the absence of the

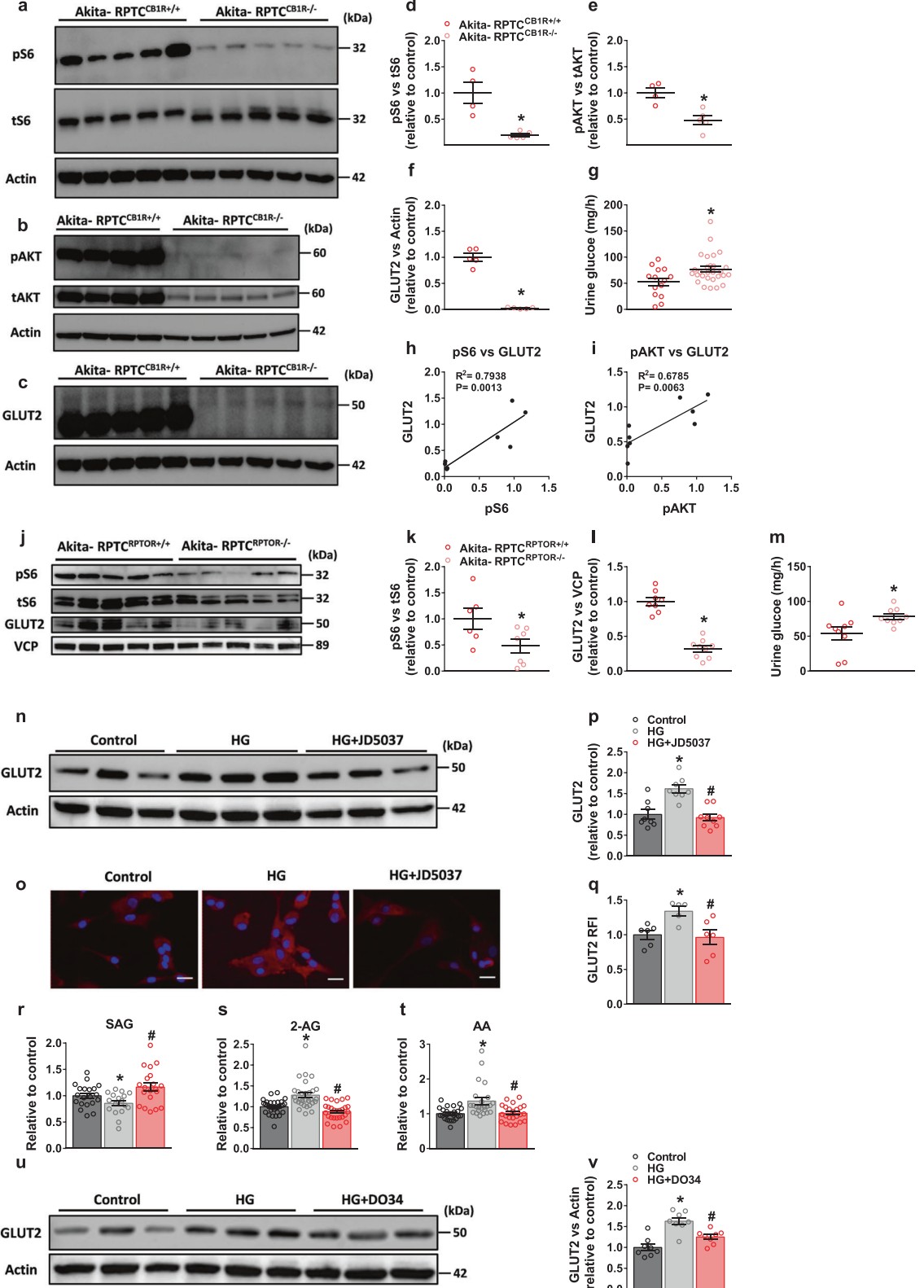

tuberous sclerosis complex (TSC) in RPTCs results in constant mTORC1 activation and elevated GLUT2 protein expression (Supplementary Fig. 2d–f), effects that are associated with tubular dysfunction, peritubular fibrosis, albuminuria, and kidney dysfunction[58]. Moreover, in diabetic mice that lack $CB_1R$, on the one hand, and have constant mTORC1 activity in their RPTCs, on the other hand (Akita-RPTC-$CB_1R^{-/-}$-$TSC^{+/-}$ animals), the beneficial effects of $CB_1R$ deletion in enhancing glucosuria, reducing proteinuria and albuminuria, and improving CCr were completely abrogated by partial deletion of TSC (Supplementary Fig. 3a–h), further supporting the regulatory role of $CB_1R$ in mTORC1 signaling and kidney homeostasis.

**Fig. 2 RPTC CB$_1$R regulates mTORC1 and GLUT2. a, d** Immunoblotting analysis and quantification of pS6 (S235/236) in kidney lysates from Akita-RPTC$^{CB1R+/+}$ and Akita-RPTC$^{CB1R-/-}$ diabetic mice. $n = 4$ mice for Akita-RPTC$^{CB1R+/+}$, $n = 5$ mice for Akita-RPTC$^{CB1R-/-}$ (*$P = 0.0032$). **b, e** Immunoblotting analysis and quantification of pAKT (S473) in kidney lysates from Akita-RPTC$^{CB1R+/+}$ and Akita-RPTC$^{CB1R-/-}$ diabetic mice. $n = 4$ mice for Akita-RPTC$^{CB1R+/+}$, $n = 5$ mice for Akita-RPTC$^{CB1R-/-}$ (*$P = 0.0056$). **c, f** Immunoblotting analysis and quantification of GLUT2 in kidney lysates from Akita-RPTC$^{CB1R+/+}$ and Akita-RPTC$^{CB1R-/-}$ diabetic mice. $n = 5$ mice per group (*$P < 0.0001$). **g** Urinary glucose levels in diabetic mice. $n = 14$ mice for Akita-RPTC$^{CB1R+/+}$, $n = 28$ mice for Akita-RPTC$^{CB1R-/-}$ (*$P = 0.0123$). **h, i** Correlation analysis between pS6 or pAKT and GLUT2 protein expression levels (**h**, $P = 0.0013$; **i**, $P = 0.0063$). **j** Representative immunoblots for pS6 and GLUT2 in kidney lysates from Akita-RPTC$^{RPTOR+/+}$ and Akita-RPTC$^{RPTOR-/-}$ diabetic mice. $n = 5$ mice per group. **k** Immunoblotting analysis for pS6 in kidney lysates. $n = 6$ mice for Akita-RPTC$^{RPTOR+/+}$, $n = 7$ mice for Akita-RPTC$^{RPTOR-/-}$ (*$P = 0.0475$). **l** Immunoblotting analysis for GLUT2 in kidney lysates. $n = 8$ mice for Akita-RPTC$^{RPTOR+/+}$, $n = 9$ mice for Akita-RPTC$^{RPTOR-/-}$ (*$P < 0.0001$). **m** Urinary glucose levels in Akita-RPTC$^{RPTOR+/+}$ and Akita-RPTC$^{RPTOR-/-}$ diabetic mice. $n = 9$ mice per group (*$P = 0.0271$). **n, p** Representative immunoblots of GLUT2 in hRPTCs treated with HG (30 mM) or HG + JD5037 (100 nM) for 24 h. $n = 8$ for Control and HG groups, $n = 9$ for HG + JD5037 (*$P = 0.0010$, #$P < 0.0001$). **o, q** Representative GLUT2 immunofluorescence staining of primary hRPTCs treated with HG (30 mM) or HG + JD5037 (100 nM) for 24 h and quantification. 10× magnification, scale bar: 100 μm. $n = 6$ for Control and HG + JD5037 groups, $n = 5$ for HG group (*$P = 0.0049$, #$P = 0.0189$). **r** LC-MS/MS quantification of SAG in primary hRPTCs treated with HG (30 mM) or HG + JD5037 (100 nM) for 1 h. $n = 20$ biological replicates for the control group, $n = 17$ for HG group, $n = 20$ for HG + JD group (*$P = 0.0467$, #$P = 0.0025$). **s** LC-MS/MS quantification of 2-AG in primary hRPTCs treated with HG (30 mM) or HG + JD5037 (100 nM) for 1 h. $n = 31$ for Control, $n = 30$ for HG group, $n = 27$ for HG + JD group (*$P < 0.0001$, #$P < 0.0001$). **t** LC-MS/MS quantification of AA in primary hRPTCs treated with HG (30 mM) or HG + JD5037 (100 nM) for 1 h. $n = 24$ for Control and HG groups, $n = 23$ for HG + JD group (*$P = 0.0010$, #$P = 0.0026$). **u, v** Representative immunoblots of GLUT2 in hRPTCs treated with HG (30 mM) or HG + DO34 (100 nM) for 24 h. $n = 8$ per group (*$P < 0.0001$, #$P = 0.0025$). Data represent the mean ± SEM and were analyzed by Unpaired Two-tailed Student's t-test or one-way ANOVA followed by Tukey test (one-sided). In **b–j** *$P < 0.05$ relative to the corresponding Akita-RPTC$^{CB1R+/+}$ or Akita-RPTC$^{RPTOR+/+}$ control groups. In **l–q** *$P < 0.05$ relative to the corresponding control group. #$P < 0.05$ relative to the HG-treated group. Source data are provided as a Source Data file.

**Hyperglycemia increases endocannabinoid 'tone' in RPTCs.** Next, we aimed to determine whether acute exposure of human primary RPTCs to hyperglycemic conditions induces an elevation in GLUT2 expression in a CB$_1$R-dependent manner. Indeed, GLUT2 protein levels were upregulated by the high glucose (HG) levels and normalized with CB$_1$R antagonism (Fig. 2n–q). The HG-induced increase in GLUT2 expression was associated with enhanced 2-arachidonoylglycerol (2-AG) synthesis, manifested by reduced levels of the precursor 1-stearoyl-2-arachidonoyl-*sn*-glycerol (SAG) and elevated 2-AG and arachidonic acid (AA) levels (Fig. 2r–t and Supplementary Fig. 4a–f). However, no changes were found in the expression of the two isoforms of diacylglycerol lipase (DAGL), the rate-limiting enzyme in the biosynthesis of 2-AG (Supplementary Fig. 5a–e). In fact, inhibiting its activity by DO34 in HG-treated hRPTCs normalized the elevated GLUT2 protein levels (Fig. 2u, v), suggesting a vicious cycle by which hyperglycemia induces CB$_1$R reactivation by 2-AG production and subsequently increases GLUT2 expression through activation of the CB$_1$R/mTORC1 signaling pathway.

**CB$_1$R modulates mTORC1 via a Gi-PI3K-dependent signaling pathway.** As a G protein-coupled receptor (GPCR), CB$_1$R may affect mTORC1 activity and consequently, GLUT2 via several cellular signaling pathways. In fact, CB$_1$R-induced phosphorylation of AKT is known to be regulated upstream via the activation (phosphorylation) of PIP2 by phosphoinositide 3-kinases (PI3K) induced through the βγ subunit of GPCR[59–61]. Investigating the effect of CB$_1$R activation either by the synthetic cannabinoid arachidonyl-2′-chloroethylamide (ACEA) or by hyperglycemia in hRPTCs revealed that both conditions increased pAKT and pS6 in a PI3K-dependent manner, since their effects were completely prevented by the PI3K inhibitor wortmannin and partially by the CB$_1$R inverse agonist JD5037 (Fig. 3a–f). CB$_1$R G-protein coupling may be biased according to the means of its activation; it stabilizes its conformation either towards Gi- or Gq-coupled signaling[62,63]. Therefore, we next examined which of the two possible molecular events is triggered by CB$_1$R agonism and hyperglycemia. We found that pretreatment of RPTCs with the Gi inhibitor, pertussis toxin (PTX), but not with the Gq inhibitor, YM254890, abolished the effect of CB$_1$R activation or HG on mTORC1 activation (Fig. 3g–l). Collectively, these results suggest

that CB$_1$R activates mTORC1 and increases GLUT2 expression in Gi- and PI3K-dependent manner.

**GLUT2 transcription is directly regulated by CB$_1$R and mTORC1.** Next, we determined the molecular mechanism by which CB$_1$R/mTORC1 axis affects GLUT2. To that end, we cloned the hGLUT2 promoter into a *Firefly* reporter plasmid (pGL3-GLUT2), which was then transfected into HEK-293 cells together with a *Renilla* plasmid for the purpose of normalization (T2; Supplementary Fig. 6a–c). Exposing these cells to HG resulted in elevated transcription of GLUT2, indicated by the increased luminescence relative response ratio (RRR; Fig. 4a, Supplementary Fig. 6d). To further evaluate the direct effect of CB$_1$R on GLUT2 transcription, we utilized the CB$_1$R-TK-d64 plasmid, in which a CB$_1$R lacking the first 64 amino acids of the long N-terminal tail[64] is expressed under the human herpes virus tyrosine kinase (TK) promoter (Supplementary Fig. 6b, c, e, f). First, we validated that CB$_1$R protein expression levels are not downregulated by co-transfecting the cells with pGL3-P1 and *Renilla* plasmids (Supplementary Fig. 6g, h). Then, we found that co-transfection of pGL3-GLUT2 with CB$_1$R-TK-d64 (T3) further enhanced GLUT2 transcription (Fig. 4b). In addition, blockade of CB$_1$R (by JD5037) normalized ACEA- or HG-induced upregulation in GLUT2 transcription (Fig. 4c, d). Similar findings were obtained in pGL3-GLUT2 and CB$_1$R-TK-d64 co-transfected cells exposed to HG in the presence of the mTORC1 inhibitor rapamycin (Fig. 4e). Interestingly, mTORC1 inhibition under normoglycemic conditions enhanced GLUT2 transcription (Fig. 4e). Taken together, these results strongly demonstrate that GLUT2 transcription under hyperglycemic conditions is regulated by a CB$_1$R/mTORC1 signaling pathway.

A few transcription factors (TFs) have been linked to regulating GLUT2 transcription in the proximal tubules[65,66]; however, their regulation by CB$_1$R and mTORC1 has not yet been explored. We therefore unbiasedly screened a list of 46 potential candidate TFs (out of ~120) that were suggested by the TFBIND software to bind the hGLUT2 promoter (Supplementary Tables 3, 4). By using a promoter-binding transcription-factor profiling assay, we found that some of the TFs that were increased following the exposure of hRPTCs to HG were also competitively inhibited by adding hGLUT2 P1 promoter (Supplementary Fig. 7a, b). Of these, only

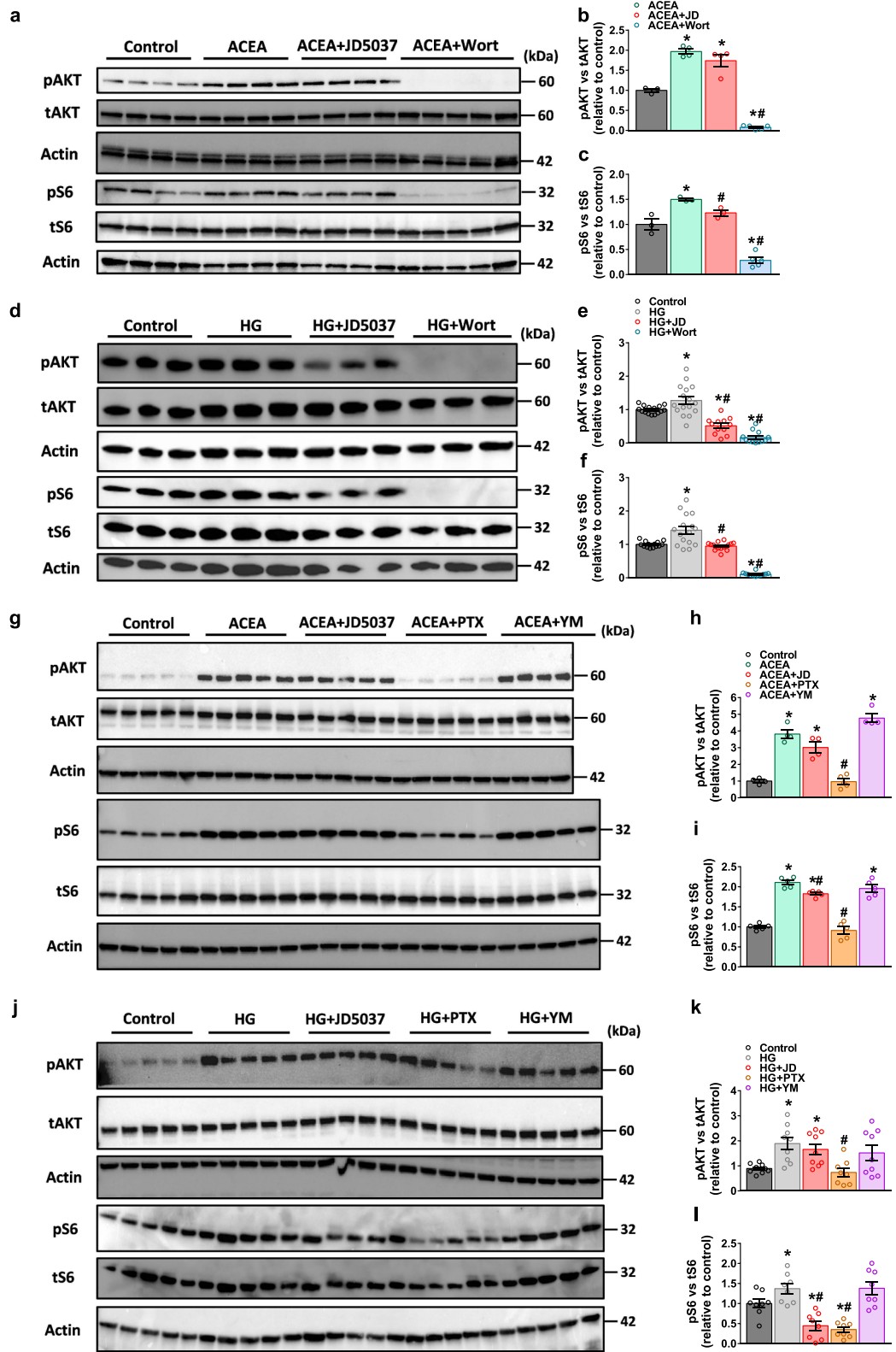

seven TFs (ATF6, STAT3, AP2, HNF4, CREB, HIF, and SREBP1) were profoundly altered (Fig. 4f), and their HG-induced activity was significantly inhibited by $CB_1R$ or mTORC1 inhibition (Fig. 4g). However, when examining their mRNA expression levels in Akita-RPTC-$CB_1R^{-/-}$ and Akita-RPTC-RPTOR$^{-/-}$ diabetic mice, only SREBP1c was decreased in the absence of $CB_1R$ or RPTOR (Fig. 4h, k), an effect that was translated to its protein

levels (Fig. 4i, j, l, m), and was found to be associated with the GLUT2 protein levels in these mice (Fig. 2c, f, j, l). Moreover, overactivation of mTORC1 in WT-RPTC-TSC$^{-/-}$ mice resulted in enhanced expression of SREBP1 and GLUT2 (Supplementary Fig. 2g, h). To directly link SREBP1 with GLUT2 transcription, we knocked down SREBP1 in hRPTCs using siRNA, whose inhibition significantly reduced the GLUT2 expression levels under HG

**Fig. 3 CB₁R modulates mTORC1 via a Gi-PI3K-dependent signaling pathway. a–c** Representative immunoblots for pAKT and pS6 in primary hRPTCs treated with or without ACEA (10 μM), ACEA + JD5037 (100 nM) or ACEA + Wortmannin (500 nM) for 1 h. For **b**, $n = 3$ for Control group, $n = 4$ for ACEA and ACEA + JD5037 groups, $n = 5$ for ACEA + Wort group. For **c**, $n = 3$ for Control, ACEA and ACEA + JD5037 groups, $n = 5$ for ACEA + Wort group (*$P < 0.0116$, #$P < 0.0141$). **d–f** Representative immunoblots for pAKT and pS6 in primary hRPTCs treated with or without HG (30 mM), HG + JD5037 (100 nM) or HG + Wortmannin (500 nM) for 1 h. For **e**, $n = 15$ for Control group, $n = 16$ for HG group and $n = 12$ for HG + JD5037 and HG + Wort groups. For **f**, $n = 15$ for Control group, $n = 16$ for HG group and $n = 12$ for HG + JD5037 and HG + Wort groups (*$P < 0.0307$, #$P < 0.0003$). **g–i** Representative immunoblots for pAKT and pS6 in primary hRPTCs treated with or without ACEA (10 μM), ACEA + JD5037 (100 nM), ACEA + Pertussis toxin (100 ng/mL) or ACEA + YM254890 (1 μM) for 1 h. For **h**, $n = 4$ for Control, ACEA, ACEA + JD and ACEA + YM groups, $n = 5$ for ACEA + PTX group. For **i**, $n = 8$ per group (*$P < 0.0009$, #$P < 0.0025$). **j–l** Representative immunoblots for pAKT and pS6 in primary hRPTCs treated with or without HG (30 mM), HG + JD5037 (100 nM), HG + Pertussis toxin (100 ng/mL) or HG + YM254890 (1 μM) for 1 h. For **k**, $n = 9$ for Control, HG, HG + JD and HG + YM groups, $n = 8$ for ACEA + PTX group. For **l**, $n = 8$ per group (*$P < 0.0422$, #$P < 0.0020$). Data represent the mean ± SEM and were analyzed by One-way ANOVA followed by Tukey test (one-sided). *$P < 0.05$ relative to the corresponding control group. #$P < 0.05$ relative to the ACEA- or HG-treated group. Source data are provided as a Source Data file.

conditions, resulting in a positive correlation between the TF and its target protein (Fig. 4n–q). In addition, acute exposure of hRPTCs to HG conditions significantly elevated the gene expression levels of SREBP1c in a CB₁R-dependent manner (Fig. 4r), and enhanced its protein translocation from the cytoplasmic compartment to the nucleus, effects that were blocked by either JD5037 or rapamycin (Fig. 4s–v). Taken together, these results imply that activation of the CB₁R/mTORC1 signaling pathway under hyperglycemic conditions regulates GLUT2 transcription via SREBP1c.

**CB₁R interacts with mTORC1 to maintain cellular homeostasis under normoglycemic conditions.** To further assess the link between CB₁R and mTORC1 in RPTCs, we followed their association under normoglycemic conditions, and surprisingly, we noted the robust activation of mTORC1 in non-diabetic WT-RPTC-CB₁R⁻/⁻ mice, as manifested by the elevated pS6 and pAKT protein levels (Fig. 5a–d). This effect was accompanied by a significant elevation in the SREBP1 and GLUT2 protein expression levels (Fig. 5e–g), indicating a strong link between CB₁R and mTORC1 and their impact on GLUT2 transcription also during healthy conditions. Whereas reduced mTORC1 activity in normoglycemic WT-RPTC-RPTOR⁻/⁻ resulted in reduced SREBP1 expression, it did not yield a significant effect on the GLUT2 levels (Fig. 5h–k).

These surprising findings of an opposite regulation of mTORC1 by CB₁R in normal conditions led us to investigate the nutritional mediators (other than glucose) that may explain how CB₁R affects mTORC1 signaling. As it is already known, mTORC1 can be activated via signals coming from branched-chain amino acids (BCAAs) and protein degradation products[67]. Indeed, CB₁R deletion in RPTCs resulted in an upregulation in the kidney mRNA and protein levels of megalin (LRP2) (Fig. 5l–o), a cellular membrane transporter responsible for the reuptake of lipoproteins, amino acids, vitamin-binding proteins, and hormones, as well as to a significant increase in the Na⁺-dependent neutral amino acid transporter B(0)AT1 (*SLC6A19*) (Fig. 5p, r, s) along with a marked upregulation in the expression of the large neutral amino acid transporter LAT1 (*SLC7A5*) (Fig. 5p, r, s). No significant changes were detected in the BCAA degrading enzymes (Fig. 5q). Interestingly, these changes were accompanied by elevated levels of kidney amino acids in WT-RPTC-CB₁R⁻/⁻ mice (Fig. 5t and Supplementary Table 7), which also persisted in the diabetic Akita-RPTC-CB₁R⁻/⁻ mice (Supplementary Fig. 8 and Supplementary Table 7), emphasizing the novel role of CB₁R in modulating amino acid uptake. SLC6A19 amino acids transporter is Na⁺ dependent, and indeed we found decreased urinary levels of Na⁺ (Fig. 5u), implicating a higher Na⁺ utilization in the absence of CB₁R in RPTCs. Similarly, in hRPTCs under normoglycemic conditions, the CB₁R antagonist JD5037 upregulated the protein

levels of SLC6A19 and SLC7A5 (Supplementary Fig. 9a–c), further enhancing amino acid bioavailability to activate mTORC1 (Supplementary Fig. 9d, e). Moreover, CB₁R antagonism enhanced BCAAs uptake in hRPTCs (Supplementary Fig. 9f–i). The overactivation of kidney mTORC1 in WT-RPTC-CB₁R⁻/⁻ mice was accompanied by increased albuminuria and kidney injury (Fig. 6a–e), evident by abnormal morphological alterations, such as enlarged glomerular and Bowman's space cross sectional areas (Fig. 6f–i). Taken together, these findings indicate that under normal conditions RPTC CB₁R could restrain mTORC1 activity via regulating nutrient absorption to maintain normal kidney morphology and function.

**Genetic reduction of GLUT2 in RPTCs protects mice from developing DKD.** Following our finding that the CB₁R/mTORC1/SREBP1c signaling pathway regulates GLUT2 transcriptional expression in RPTCs, we next assessed whether a reduction in GLUT2 expression is sufficient to prevent DKD. To that end, we utilized the same breeding paradigm described earlier to generate diabetic mice lacking GLUT2 in RPTCs (Akita-RPTC-GLUT2⁻/⁻) and their littermate control mice (Akita-RPTC-GLUT2⁺/⁺) (Supplementary Fig. 10a–e). Whereas genetic reduction of GLUT2 specifically in RPTCs did not affect the susceptibility of the mice to develop insulin deficient diabetes (Fig. 7a–e), the null mice exhibited increased glycosuria (Fig. 7f) and preserved kidney function, manifested by reduced kidney-to-body weight ratio, urine excretion-to-water consumption ratio, as well as reduced proteinuria, albuminuria, urine creatinine, ACR, and urinary KIM-1 (Fig. 7g–m), without affecting BUN or CCr (Fig. 7n, o). Moreover, kidney morphological anomalies were prevented, including glomerular and Bowman's space areas hypertrophy and mesangial expansion (Fig. 7p–s). The expression of kidney injury, fibrosis, and inflammation markers was reduced in Akita-RPTC-GLUT2⁻/⁻ mice (Fig. 7t–v), suggesting that GLUT2 plays an obligatory role in the development of DKD. To further determine the role of RPTC-GLUT2 in glucose reabsorption by the kidney under diabetic conditions, we utilized an in vivo PET-MRI assay in which the GLUT high-affinity substrate, 2-deoxy-2-[¹⁸F]-fluoro-D–glucose (FDG), was injected into the tail vein and its uptake by different organs was analyzed. Kidney [¹⁸F]-FDG uptake was significantly increased in the diabetic mice (Fig. 7w–x), consequently increasing its concentration in the circulation and uptake in peripheral organs (e.g., the liver and muscle) while reducing its accumulation in the bladder (Supplementary Fig. 11a–h). Notably, these alterations were completely reversed in Akita-RPTC-GLUT2⁻/⁻ mice (Fig. 7w–x and Supplementary Fig. 11a–h), indicating the key role played by RPTC-GLUT2 in kidney glucose reabsorption. Interestingly, we found that deletion of GLUT2 in RPTCs significantly

T2- pGL3-GLUT2+ Renilla
T3- pGL3-GLUT2+ Renilla+ CB₁R TK d64

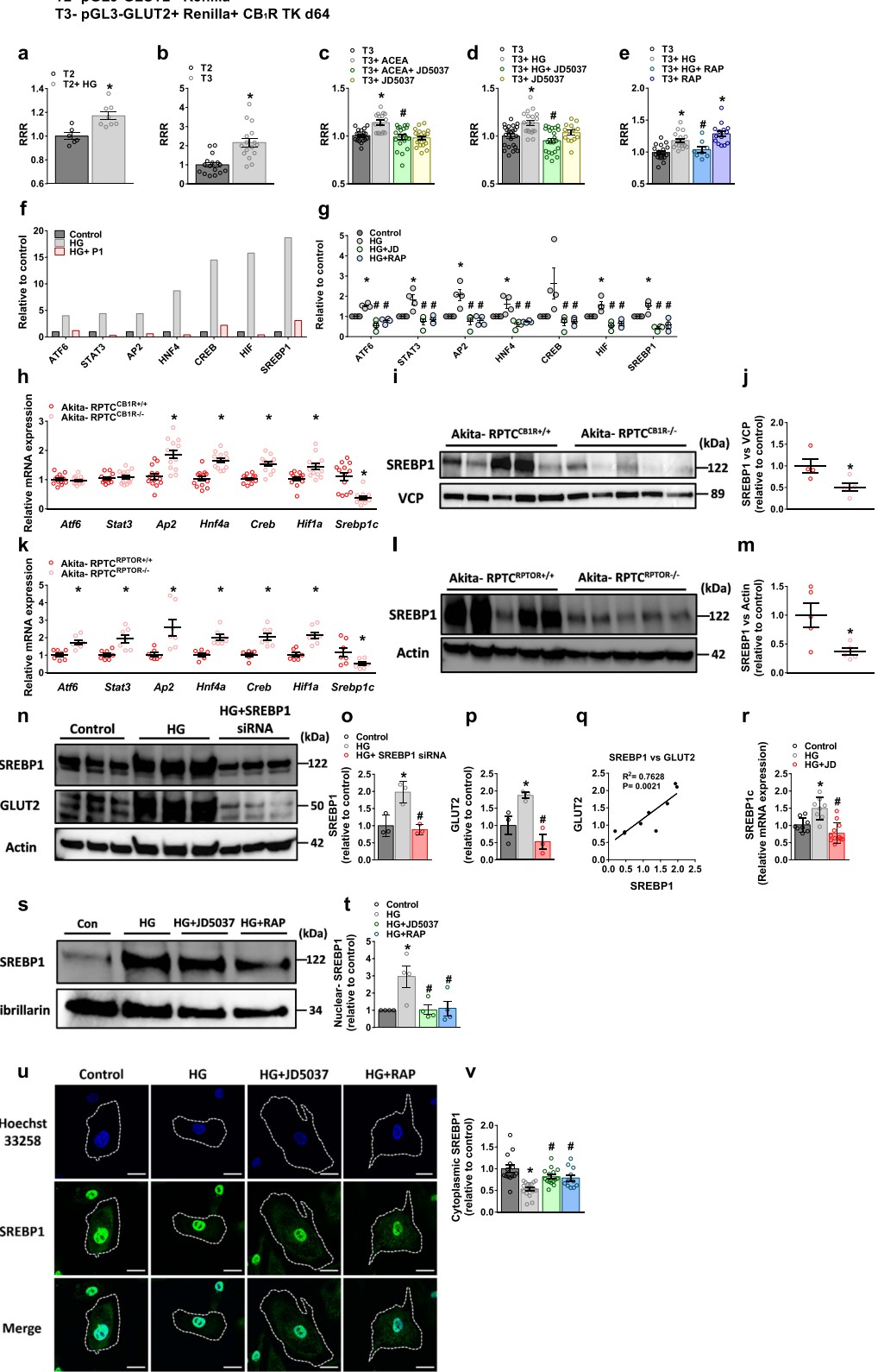

downregulated SGLT2 expression under both normo- and hyperglycemic conditions in mice (Supplementary Fig. 12a–d). These findings may suggest that GLUT2 has a central role in modulating kidney glucose reabsorption and homeostasis by governing glucose transporters in the proximal tubules.

## Discussion

The present study reveals, for the first time, the existence of a CB₁R/mTORC1 signaling axis in RPTCs and its importance in regulating kidney function in health and disease. Whereas CB₁R may restrain potentially deleterious mTORC1 over-activation

**Fig. 4 GLUT2 transcription is directly regulated by CB$_1$R and mTORC1. a–e** Luminescence relative response ratio (RRR) analyses of HEK293 cells co-transfected transiently with pGL3-GLUT and *Renilla* luciferase plasmids (T2) or with pGL3-GLUT, *Renilla* luciferase, and CB$_1$R-TK-d64 plasmids (T3), treated with or without ACEA (10 μM; **b**, **c**) or with HG (30 mM; **a**, **d**, **e**) for 3 h, in the presence or absence of JD5037 (100 nM) or rapamycin (100 nM). For **a**, $n = 6$ for T2, $n = 8$ for T2+HG (*$P = 0.0034$). For **b**, $n = 16$ per group (*$P < 0.0001$). For **c**, $n = 19$ for T3 group, $n = 16$ for T3 + ACEA group and $n = 20$ for T3 + ACEA + JD5037 and T3 + JD5037 groups (*$P < 0.0001$, #$P = 0.0002$). For **d**, $n = 23$ for T3 and T3 + HG + JD5037 groups, $n = 19$ for T3 + HG group and $n = 14$ for T3 + JD5037 group (*$P = 0.0003$, #$P = 0.0002$). For **e**, $n = 18$ for T3 group, $n = 16$ for T3 + HG group, $n = 8$ for T3 + HG + RAP group and $n = 14$ for T3 + RAP group (*$P < 0.0001$, #$P = 0.0207$). **f** Promoter-binding transcription-factor (TF) profiling assay of the eight most prominent candidate TFs present in nuclear extracts (NEs) collected from primary hRPTCs treated with or without HG (30 mM) for 3 h in the presence or absence of hGLUT2 promoter (P1). **g** Transcription-factor profiling assay for eight candidate TFs present at the NEs of primary hRPTCs treated with or without HG (30 mM) in the presence or absence of JD5037 (100 nM) or rapamycin (100 nM) for 3 h. $n = 4$ per group (*$P < 0.0019$, #$P < 0.0001$). **h** qPCR analyses of kidney Glut2 candidate TFs in kidney lysate from Akita-RPTC$^{CB1R+/+}$ and Akita-RPTC$^{CB1R-/-}$ diabetic mice. $n = 13$ mice per group (*$P < 0.0018$). **i, j** Immunoblotting analysis and quantification of SREBP1 in kidney lysate from Akita-RPTC$^{CB1R+/+}$ and Akita-RPTC$^{CB1R-/-}$ diabetic mice. $n = 4$ mice for Akita-RPTC$^{CB1R+/+}$, $n = 5$ mice for Akita-RPTC$^{CB1R-/-}$ (*$P = 0.0264$). **k** qPCR analysis and quantification of kidney Glut2 candidate TFs in kidney lysate from Akita-RPTC$^{RPTOR+/+}$ and Akita-RPTC$^{RPTOR-/-}$ diabetic mice. $n = 7$ mice per group (*$P < 0.0324$). **l, m** Immunoblotting analysis and quantification of SREBP1 in kidney lysate from Akita-RPTC$^{RPTOR+/+}$ and Akita-RPTC$^{RPTOR-/-}$ diabetic mice. $n = 5$ mice per group (*$P = 0.0191$). **n–q** Immunoblotting analysis of SREBP1c and GLUT2 in primary hRPTCs treated with or without HG (30 mM) in the presence or absence of SREBP1 siRNA for 24 h. $n = 3$ biological replicates per group (*$P < 0.0385$, #$P < 0.0051$). **r** qPCR analysis of SREBP1c in primary hRPTCs treated with or without HG (30 mM) in the presence or absence of JD5037 (100 nM) for 1 h. $n = 8$ for Control and HG groups, $n = 12$ for HG + JD group (*$P = 0.0031$, #$P < 0.0001$). **s, t** Representative immunoblotting analysis and quantification of nuclear SREBP1 in extracts collected from primary hRPTCs treated with or without HG (30 mM) in the presence or absence of JD5037 (100 nM) or rapamycin (100 nM) for 3 h. SREBP1 expression was normalized to the nuclear marker Fibrillarin. $n = 4$ per group (*$P = 0.0196$, #$P < 0.0497$). **u, v** Representative immunofluorescence and quantification of cytoplasmic SREBP1 expression in primary hRPTCs treated with or without HG (30 mM) in the presence or absence of JD5037 (100 nM) or rapamycin (100 nM) for 3 h. 100× magnification, scale bar: 10 μm. $n = 14$ for Control group, $n = 19$ for HG group, $n = 15$ for HG + JD5037 group and $n = 11$ for HG + RAP group (*$P < 0.0001$, #$P < 0.0012$). Data represent the mean ± SEM and were analyzed by unpaired two-tailed Student's t-test or one-way ANOVA followed by Tukey test (one-sided). *$P < 0.05$ relative to the corresponding control group. #$P < 0.05$ relative to the ACEA- or HG-treated group. Source data are provided as a Source Data file.

during normoglycemia by preventing amino acid flux, in diabetes, enhanced activity of CB$_1$R by eCBs, stimulates mTORC1 to further increase glucose uptake via upregulating SREBP1-mediated GLUT2 expression, thus contributing to the development of DKD (Fig. 8). These novel findings have developmental, functional, and translational implications.

Under normoglycemic physiological condition, kidney CB$_1$R regulates hemodynamic by inducing afferent arterioles vasodilation and reducing GFR[68,69]. CB$_1$R has been suggested to regulate Na$^+$/K$^+$-ATPase activity in RPTCs[63,70], which is required for maintaining the electrochemical sodium gradient across the brush border membrane of the RPTCs, and it is therefore essential for the reabsorption process that occurs in the kidney. Similar to CB$_1$R, mTORC1 also plays an essential physiological role in regulating nutrient transport in RPTCs[71]. Mice lacking mTORC1 in RPTCs[71] or treated with rapamycin develop glycosuria, phosphaturia, aminoaciduria, low-molecular weight proteinuria, and albuminuria[72], findings observed in individuals suffering from the Fanconi-Bickel syndrome, caused by mutations in the GLUT2 gene, *SLC2A2*[73,74]. Interestingly, we found that deletion of CB$_1$R from RPTCs in non-diabetic animals stimulated mTORC1 signaling resulting in kidney dysfunction. This effect was associated with increased amino acid content and bioavailability in RPTC-CB$_1$R$^{-/-}$ mice and hRPTCs treated with a CB$_1$R antagonist. Increased BCAA content in the kidney of these animals and cells is most likely mediated via megalin and the neutral amino acid transporter SLC6A19 as well as the large neutral amino acid transporter, SLC7A5. CB$_1$R has been previously shown to regulate kidney megalin expression in diet-induced obesity[75,76]. SLC6A19 was also suggested as a potential target for the treatment of metabolic disorders, since its absence in mice was associated with decreased mTORC1 activity[77,78]. In fact, the absence of CB$_1$R may alter Na$^+$/K$^+$-ATPase[63,70], required for SLC6A19 activity. SLC7A5, known to be expressed in proximal tubular cells[79], was shown to activate mTORC1 via transporting Leucine into the lysosome[80]. We show here that these transporters are regulated by CB$_1$R. Although Cystine levels were highly upregulated in mice lacking CB$_1$R, no changes were detected in the levels of the Cystine transporter SLC3A1. Grahammer and

colleagues showed that genetic depletion of mTORC1 in the proximal tubules decreased the expression and/or phosphorylation of multiple amino acid transporters, leading to aminoaciduria[71], emphasizing a direct role for mTORC1 in amino acid transport. We have previously reported that genetic manipulations of mTORC1 in RPTCs, both constant activation, and inhibition by TSC1 KO and RPTOR KO respectively, modulate SLC6A19 expression[58]. Therefore, we cannot exclude the possibility that prolonged mTORC1 activation in RPTC-CB$_1$R$^{-/-}$ mice per se lead to enhanced amino acid transport. Nevertheless, megalin and cubilin expression was unchanged in mTORC1-depleted tubules[71]; therefore, the specific role of CB$_1$R in modulating mTORC1 activity via megalin/cubilin mediated amino acids transport cannot be negated. Moreover, we previously reported that genetic deletion of CB$_1$R in RPTCs increases kidney weight and glomerular area[21], implying that CB$_1$R may play a developmental role in the kidney. The present study offers an explanation for this phenomenon through a hitherto unrecognized CB$_1$R-mediated regulation of mTORC1, known to control cell metabolism, growth, and proliferation. Thus, our current findings raise a concern for using CB$_1$R antagonists in normoglycemic patients because they may enhance mTORC1 activity and induce deleterious effects on the kidney.

Whereas under normoglycemic conditions CB$_1$R controls mTORC1 activation in RPTCs via restraining amino acids flux, our findings demonstrate that glucotoxicity enhances CB$_1$R-induced mTORC1 activity in the same type of cells. In accordance with previous findings with CB$_1$R blockers[17,19,25], specific deletion of RPTC-CB$_1$R in *Akita* diabetic mice, although not affecting the diabetic phenotype of the animals, had a profound beneficial effect on the kidney, which can be attributed to the inhibition of mTORC1 signaling. These findings were associated with reduced GLUT2 expression and glucosuria, effects that were previously described by us as being modulated by kidney CB$_1$R[25]. In agreement with our findings, it was recently demonstrated that pancreatic GLUT2 levels are reduced in CB$_1$R/CB$_2$R-deficient mice[81]. Mechanistically, our findings show here that CB$_1$R regulates mTORC1 in RPTCs via the PI3K-Akt pathway under diabetic

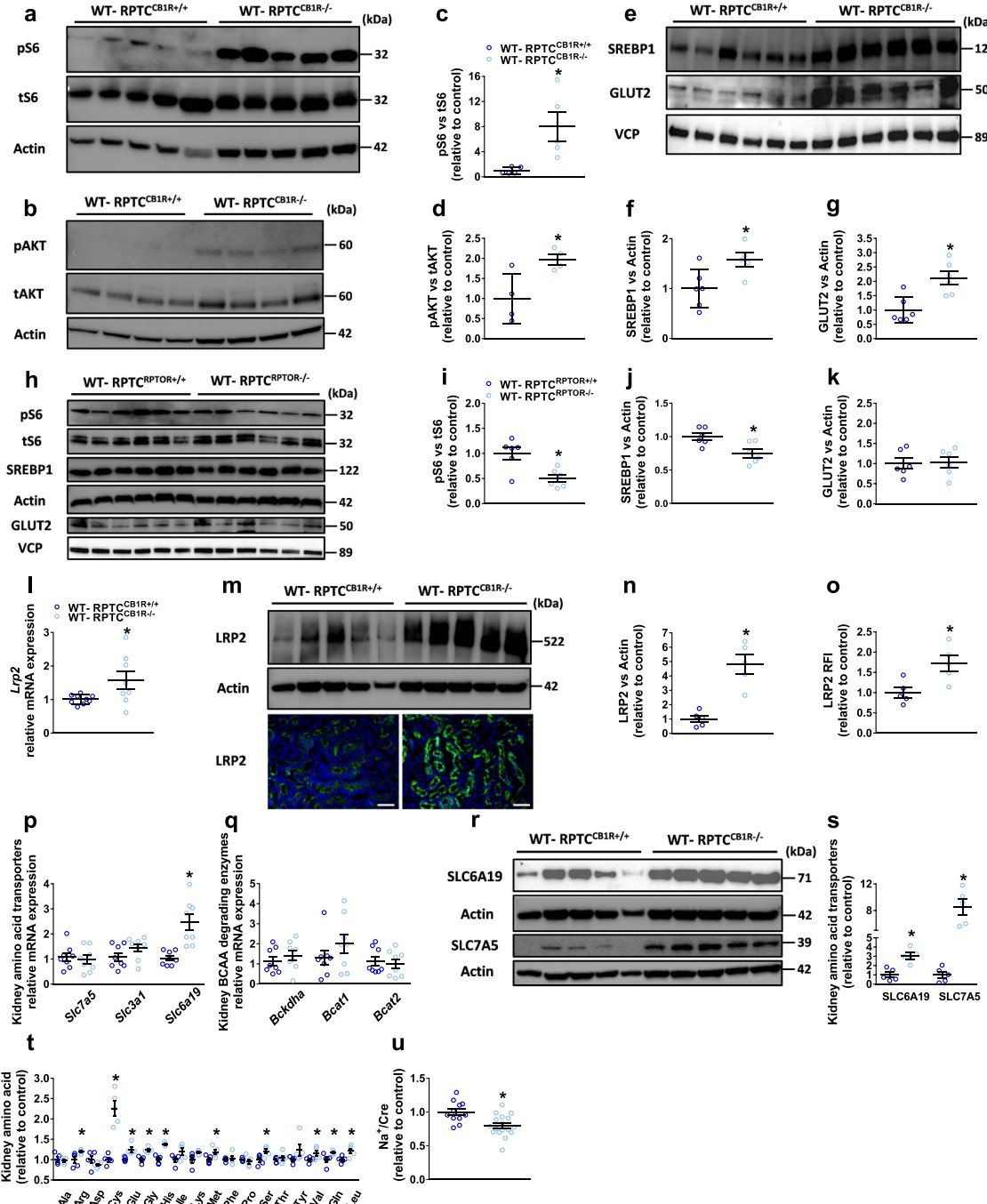

conditions. Dalton and colleagues suggested that the βγ subunits of the $CB_1R$-coupled $G_{i/o}$ protein in neuronal cells stimulate PI3K to induce ERK phosphorylation[59]. On the contrary, we found that stimulation of $CB_1R$ by glucose did not affect (or even reduced) the phosphorylation of ERK in RPTCs and that mTORC1 activation is mediated via a $G_{i/o}$ protein coupled $CB_1R$ pathway. We and others[82,83] have previously shown that $CB_1R$ may also signal via $G_q$ signaling, which enhances cellular calcium influx, contributing to increased PKC-β1 expression, resulting in dynamic translocation of GLUT2 into the brush border membrane of the RPTCs[25]. Such a duality in the cellular signaling cascades modulated by $CB_1R$ can exist in RPTCs[63], allowing $CB_1R$ to modulate both the expression and the translocation of GLUT2. Alternatively, $CB_1R$ activation can stimulate phospholipase C (PLC) via the $G_{i/o}$ βγ subunit, thereby increasing intracellular $[Ca^{2+}]$ influx and PKC

activation[61]. Consistently, we found (i) significant positive correlation between pS6 and pAKT and GLUT2, (ii) modulation of GLUT2 expression in mice lacking RPTOR or TSC1 specifically in RPTCs, and (iii) increased transcriptional expression of GLUT2 by HG and/or $CB_1R$ activation, effects that were completely prevented by $CB_1R$ or mTORC1 inhibition.

As per the transcriptional regulation of GLUT2 in RPTCs[65,66], we found 24 TFs that were enriched in the nuclear compartment of hRPTCs exposed to HG and that selectively bind to the GLUT2 promoter. The most prominent one, SREBP1c, has been previously shown to promote kidney lipotoxicity[84–87]. It has been also reported that in RPTCs, hepatocytes, and Schwann cells, HG stimulates SREBP1c via the PI3K/AKT/mTORC1 pathway[42,87–90]. Moreover, SREBP1c enhances GLUT2 expression in hepatocytes exposed to

**Fig. 5 CB₁R regulates mTORC1 activation under normoglycemic conditions.** **a**, **c** Immunoblotting analysis and quantification of pS6 in kidney lysates from WT-RPTC^CB1R+/+ and WT-RPTC^CB1R−/− normoglycemic mice. $n = 5$ mice per group (*$P = 0.0166$). **b**, **d** Immunoblotting analysis and quantification of pAKT in kidney lysates from WT-RPTC^CB1R+/+ and WT-RPTC^CB1R−/− normoglycemic mice. $n = 4$ mice per group (*$P = 0.0275$). **e**–**g** Immunoblotting analysis and quantification of SREBP1c and GLUT2 in kidney lysates from WT-RPTC^CB1R+/+ and WT-RPTC^CB1R−/− normoglycemic mice. $n = 6$ mice per group (*$P < 0.0264$). **h**–**k** Immunoblotting analysis and quantification of pS6, SREBP1c, and GLUT2 in kidney lysates from WT-RPTC^RPTOR+/+ and WT-RPTC^RPTOR−/− normoglycemic mice. $n = 6$ mice per group (*$P < 0.0136$). **l** qPCR analyses of megalin (*Lrp2*) in kidney lysates from WT-RPTC^CB1R+/+ and WT-RPTC^CB1R−/− normoglycemic mice. $n = 9$ mice for WT-RPTC^CB1R+/+, $n = 8$ mice for WT-RPTC^CB1R−/− (*$P = 0.0395$). **m** Representative LRP2 immunoblotting and immunofluorescence staining of kidney sections from WT-RPTC^CB1R+/+ and WT-RPTC^CB1R−/− normoglycemic mice. 10× magnification, scale bar 100 μm. $n = 5$ mice per group. **n** Immunoblotting quantification of LRP2 in kidney lysates from WT-RPTC^CB1R+/+ and WT-RPTC^CB1R−/− normoglycemic mice. $n = 5$ mice per group (*$P = 0.0007$). **o** Immunofluorescence quantification for LRP2 in kidney sections from WT-RPTC^CB1R+/+ and WT-RPTC^CB1R−/− normoglycemic mice. 10× magnification, scale bar 100 μm. $n = 5$ mice per group (*$P = 0.0164$). **p** qPCR analysis of amino acid transporters in kidney lysates from WT-RPTC^CB1R+/+ and WT-RPTC^CB1R−/− normoglycemic mice. $n = 8$ mice per group (*$P = 0.0007$). **q** qPCR analysis of BCAA degrading enzymes in kidney lysates from WT-RPTC^CB1R+/+ and WT-RPTC^CB1R−/− normoglycemic mice. $n = 8$ mice per group. **r**, **s** Immunoblotting analysis and quantification of SLC6A19 and SLC7A5 in kidney lysates from WT-RPTC^CB1R+/+ and WT-RPTC^CB1R−/− normoglycemic mice. $n = 5$ mice per group (*$P < 0.0017$). **t** LC-MS/MS quantification of amino acids in kidney lysates from WT-RPTC^CB1R+/+ and WT-RPTC^CB1R−/− normoglycemic mice. $n = 5$ mice per group (*$P < 0.0354$). **u** Urinary Na⁺ levels normalized to the creatinine concentration. $n = 11$ mice for WT-RPTC^CB1R+/+, $n = 17$ mice for WT-RPTC^CB1R−/− (*$P = 0.0031$). Data represent the mean ± SEM and were analyzed by unpaired two-tailed Student's t-test. *$P < 0.05$ relative to the corresponding control groups. Source data are provided as a Source Data file.

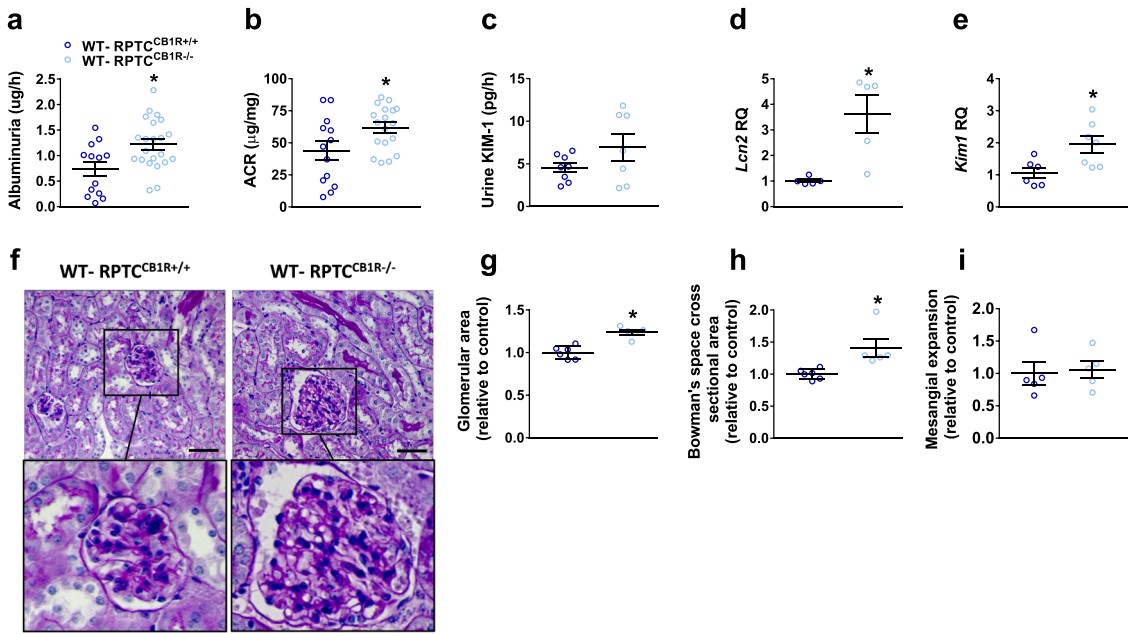

**Fig. 6 mTORC1 overactivation in RPTC-CB₁R KO mice promotes kidney functional and morphological abnormalities.** **a** Albuminuria in 16-week-old normoglycemic mice. $n = 13$ mice for WT-RPTC^CB1R+/+, $n = 22$ mice for WT-RPTC^CB1R−/− (*$P = 0.0079$). **b** Urine albumin-to-creatinine ratio (ACR) in 16-week-old normoglycemic mice. $n = 13$ mice for WT-RPTC^CB1R+/+, $n = 18$ mice for WT-RPTC^CB1R−/− (*$P = 0.0301$). **c** Urinary KIM-1 levels in 16-week-old normoglycemic mice. $n = 8$ mice for WT-RPTC^CB1R+/+, $n = 7$ mice for WT-RPTC^CB1R−/−. **d** qPCR analysis of the kidney injury marker, *Lcn2*. $n = 5$ mice per group (*$P = 0.0070$). **e** qPCR analysis of the kidney injury marker, *Kim1*. $n = 6$ mice for WT-RPTC^CB1R+/+, $n = 7$ mice for WT-RPTC^CB1R−/− (*$P = 0.0197$). **f** Representative PAS staining of the kidney, 40× magnification, scale bar: 50 μm. **g** Glomerular area quantification (at least 10 glomeruli per mouse). $n = 6$ mice for WT-RPTC^CB1R+/+, $n = 5$ mice for WT-RPTC^CB1R−/− (*$P = 0.0004$). **h** Bowman's space cross-sectional area quantification (at least 10 glomeruli per mouse). $n = 6$ mice for WT-RPTC^CB1R+/+, $n = 5$ mice for WT-RPTC^CB1R−/− (*$P = 0.0142$). **i** Mesangial expansion quantification (at least 10 glomeruli per mouse). $n = 6$ mice for WT-RPTC^CB1R+/+, $n = 5$ mice for WT-RPTC^CB1R−/−. Data represent the mean ± SEM and were analyzed by Unpaired Two-tailed Student's t-test. *$P < 0.05$ relative to the corresponding control groups. Source data are provided as a Source Data file.

hyperglycemia[91]. In line with the recent reports, we also found that specific genetic deletion of CB₁R, RPTOR, or TSC1 in the RPTCs modulated SREBP1c levels in parallel to GLUT2 expression. Furthermore, treatment of hRPTCs with rapamycin stimulated GLUT2 transcriptional activity. Collectively, these findings demonstrate the importance of SREPB1c in modulating the CB₁R/mTORC1 effect on GLUT2.

Here, we also demonstrate a direct effect of glucose on the production and/or degradation of 2-AG, which in turn, may activate its own receptor to further promote glucose transport resulting in RPTC's dysfunction. These findings are in line with previous reports showing increased eCB/CB₁R 'tone' in different compartments/cells within the kidney as well as in pancreatic islets under gluco- and/or lipo-toxic conditions[17,20,21,92]. As previously reported by Sampaio and colleagues, the two isoforms of DAGL are differentially expressed in the renal proximal tubule[63]. While we did not find any changes in their expression in response to hyperglycemia, the functional inhibition of DAGL

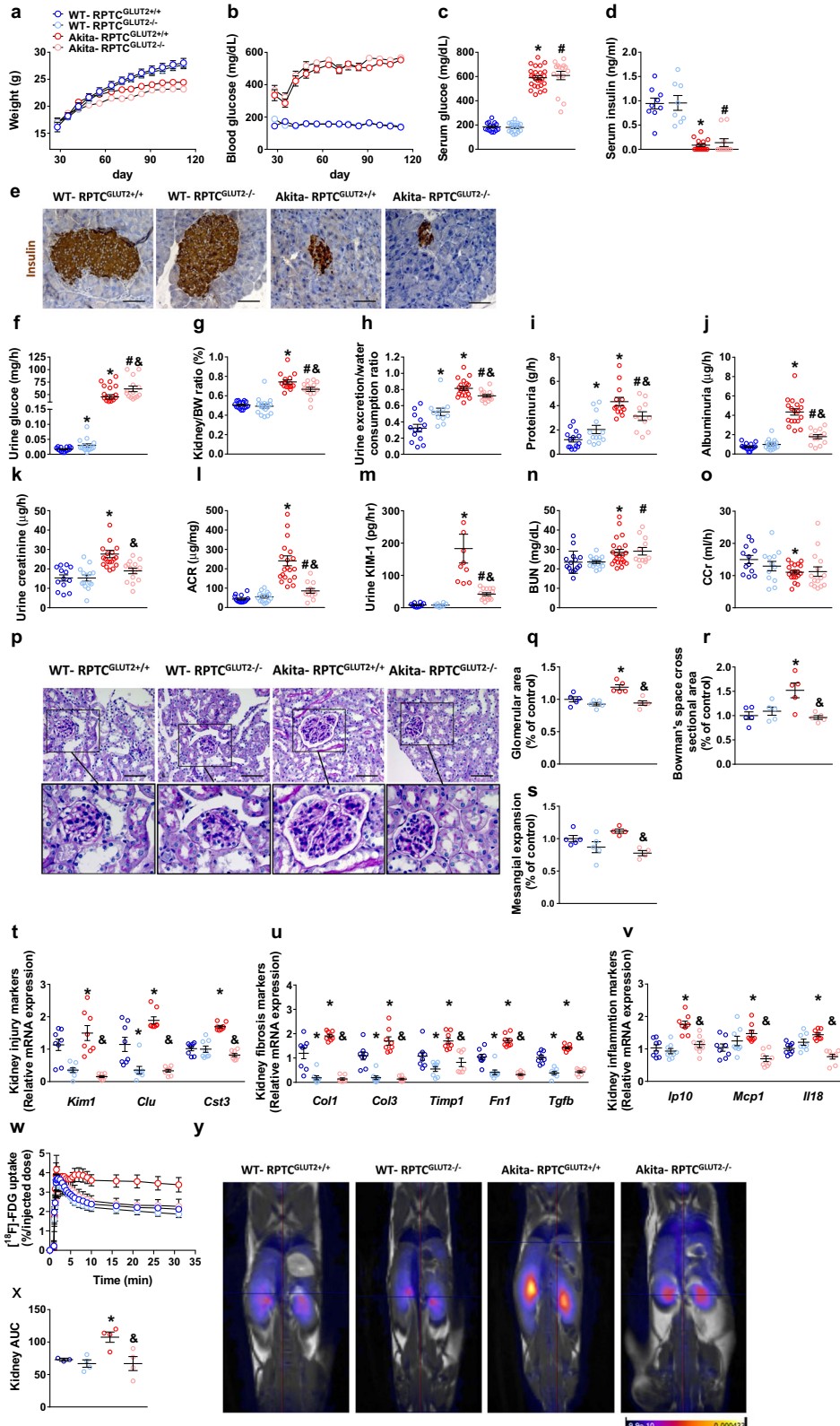

resulted in reduced GLUT2 levels in hRPTCs, suggesting that increased 2-AG levels by glucose are mediated via DAGL.

Global GLUT2$^{-/-}$ mice display severe glycosuria[93], and the complete absence of GLUT2 at the basolateral membrane of the RPTCs supposedly leads to enhanced intracellular glucose levels and proximal tubular damage, even in the absence of diabetes. This effect has been well-documented in Fanconi-Bickel Syndrome patients who exhibit tubular dysfunction[94]. However, our current study revealed additional surprising phenomena. Specific reduction, but not full ablation of GLUT2 in RPTCs ameliorates

**Fig. 7 Genetic reduction of GLUT2 in RPTCs protects mice from developing DKD.** All the following measurements were done in Akita diabetic mice and their littermate WT controls with or without reduced expression of GLUT2 in RPTCs: **a** Body weight surveillance for a period of 16 weeks. $n = 16$ mice for WT-RPTC$^{GLUT2+/+}$, $n = 17$ mice for WT-RPTC$^{GLUT2-/-}$, $n = 20$ mice for Akita-RPTC$^{GLUT2+/+}$, $n = 17$ mice for Akita-RPTC$^{GLUT2-/-}$. **b** Blood glucose surveillance for a period of 16 weeks. $n = 16$ mice for WT-RPTC$^{GLUT2+/+}$, $n = 17$ mice for WT-RPTC$^{GLUT2-/-}$, $n = 20$ mice for Akita-RPTC$^{GLUT2+/+}$, $n = 17$ mice for Akita-RPTC$^{GLUT2-/-}$. **c** Serum glucose. $n = 16$ mice for WT-RPTC$^{GLUT2+/+}$, $n = 17$ mice for WT-RPTC$^{GLUT2-/-}$, $n = 20$ mice for Akita-RPTC$^{GLUT2+/+}$, $n = 17$ mice for Akita-RPTC$^{GLUT2-/-}$ (*$P < 0.0062$, #$P = 0.0002$, &$P = 0.0082$). **d** Serum insulin. $n = 9$ mice for WT-RPTC$^{GLUT2+/+}$, $n = 8$ mice for WT-RPTC$^{GLUT2-/-}$, $n = 15$ mice for Akita-RPTC$^{GLUT2+/+}$, $n = 10$ mice for Akita-RPTC$^{GLUT2-/-}$ (*$P < 0.0437$, #$P = 0.0001$). **e** Representative insulin immunohistochemistry staining in the pancreas, 40× magnification, scale bar: 50 μm. **f** Urinary glucose levels. $n = 16$ mice for WT-RPTC$^{GLUT2+/+}$, $n = 17$ mice for WT-RPTC$^{GLUT2-/-}$, $n = 20$ mice for Akita-RPTC$^{GLUT2+/+}$, $n = 17$ mice for Akita-RPTC$^{GLUT2-/-}$ (*$P < 0.0062$, #$P < 0.0001$, &$P = 0.0288$). **g** Kidney-to-body weight ratio. $n = 16$ mice for WT-RPTC$^{GLUT2+/+}$, $n = 17$ mice for WT-RPTC$^{GLUT2-/-}$, $n = 20$ mice for Akita-RPTC$^{GLUT2+/+}$, $n = 17$ mice for Akita-RPTC$^{GLUT2-/-}$ (*$P < 0.0001$, #$P < 0.0001$, &$P = 0.0187$). **h** Urine exertion-to-water consumption ratio. $n = 16$ mice for WT-RPTC$^{GLUT2+/+}$, $n = 17$ mice for WT-RPTC$^{GLUT2-/-}$, $n = 20$ mice for Akita-RPTC$^{GLUT2+/+}$, $n = 17$ mice for Akita-RPTC$^{GLUT2-/-}$ (*$P < 0.0062$, #$P = 0.0002$, &$P = 0.0082$). **i** Urinary protein levels. $n = 14$ mice for WT-RPTC$^{GLUT2+/+}$, $n = 13$ mice for WT-RPTC$^{GLUT2-/-}$ and Akita-RPTC$^{GLUT2+/+}$, $n = 11$ mice for Akita-RPTC$^{GLUT2-/-}$ (*$P < 0.0418$, #$P = 0.0389$, &$P = 0.0282$). **j** Urinary albumin levels. $n = 15$ mice for WT-RPTC$^{GLUT2+/+}$, $n = 15$ mice for WT-RPTC$^{GLUT2-/-}$, $n = 18$ mice for Akita-RPTC$^{GLUT2+/+}$, $n = 13$ mice for Akita-RPTC$^{GLUT2-/-}$ (*$P < 0.0001$, #$P = 0.0030$, &$P < 0.0001$). **k** Urinary creatinine levels. $n = 13$ mice for WT-RPTC$^{GLUT2+/+}$, $n = 14$ mice for WT-RPTC$^{GLUT2-/-}$, $n = 20$ mice for Akita-RPTC$^{GLUT2+/+}$, and $n = 14$ mice for Akita-RPTC$^{GLUT2-/-}$ (*$P < 0.0001$, &$P = 0.0019$). **l** Urine albumin-to-creatinine ratio (ACR). $n = 14$ mice for WT-RPTC$^{GLUT2+/+}$, $n = 14$ mice for WT-RPTC$^{GLUT2-/-}$, $n = 20$ mice for Akita-RPTC$^{GLUT2+/+}$, and $n = 12$ mice for Akita-RPTC$^{GLUT2-/-}$ (*$P < 0.0001$, #$P = 0.0326$, &$P = 0.0001$). **m** Urinary KIM-1 levels. $n = 10$ mice for WT-RPTC$^{GLUT2+/+}$, $n = 8$ mice for WT-RPTC$^{GLUT2-/-}$, $n = 10$ mice for Akita-RPTC$^{GLUT2+/+}$, and $n = 14$ mice for Akita-RPTC$^{GLUT2-/-}$ (*$P = 0.0010$, #$P < 0.0001$, &$P = 0.0011$). **n** Urinary BUN. $n = 15$ mice for WT-RPTC$^{GLUT2+/+}$, $n = 15$ mice for WT-RPTC$^{GLUT2-/-}$, $n = 22$ mice for Akita-RPTC$^{GLUT2+/+}$ and $n = 12$ mice for Akita-RPTC$^{GLUT2-/-}$ (*$P = 0.0261$, #$P = 0.0076$). **o** Creatinine clearance (CCr). $n = 12$ mice for WT-RPTC$^{GLUT2+/+}$, $n = 12$ mice for WT-RPTC$^{GLUT2-/-}$, $n = 20$ mice for Akita-RPTC$^{GLUT2+/+}$, and $n = 15$ mice for Akita-RPTC$^{GLUT2-/-}$ (*$P = 0.0034$). **p** Representative PAS staining of the kidney, 40× magnification, scale bar: 50 μm. **q** Glomerular area quantification (at least 10 glomeruli per mouse). $n = 5$ mice per group (*$P = 0.0096$, &$P = 0.0015$). **r** Bowman's space cross-sectional area quantification (at least 10 glomeruli per mouse). $n = 5$ mice per group (*$P = 0.0160$, &$P = 0.0075$). **s** Mesangial expansion quantification (at least 10 glomeruli per mouse). $n = 5$ mice for WT-RPTC$^{GLUT2+/+}$ and WT-RPTC$^{GLUT2-/-}$, $n = 4$ mice for Akita-RPTC$^{GLUT2+/+}$ and Akita-RPTC$^{GLUT2-/-}$ (&$P = 0.0006$). **t** qPCR analysis of kidney injury markers *Kim1*, *Clu*, and *Cst3*. $n = 8$ mice per group (*$P < 0.0091$, &$P < 0.0001$). **u** qPCR analysis of kidney fibrogenic markers *Col1*, *Col3*, *Timp1*, *Fn1*, and *Tgfb*. $n = 8$ mice per group (*$P < 0.0173$, &$P < 0.0008$). **v** qPCR analyses of kidney inflammatory markers *Ip10*, *Mcp1*, and *Il18*. $n = 8$ mice per group (*$P < 0.0122$, &$P < 0.0006$). **w, x** Kidney uptake of [¹⁸F]-FDG using PET-MRI analysis. $n = 3$ mice for WT-RPTC$^{GLUT2+/+}$, $n = 4$ mice for WT-RPTC$^{GLUT2-/-}$, Akita-RPTC$^{GLUT2+/+}$ and Akita-RPTC$^{GLUT2-/-}$ (*$P = 0.0139$, &$P = 0.0224$). **y** Representative kidneys PET-MRI images. Data represent the mean ± SEM and were analyzed by one-way ANOVA followed by Tukey test (one-sided). *$P < 0.05$ relative to the WT-RPTC$^{GLUT2+/+}$ mice; #$P < 0.05$ relative to WT-RPTC$^{GLUT2-/-}$ mice; &$P < 0.05$ relative to the Akita-RPTC$^{GLUT2+/+}$ mice. Source data are provided as a Source Data file.

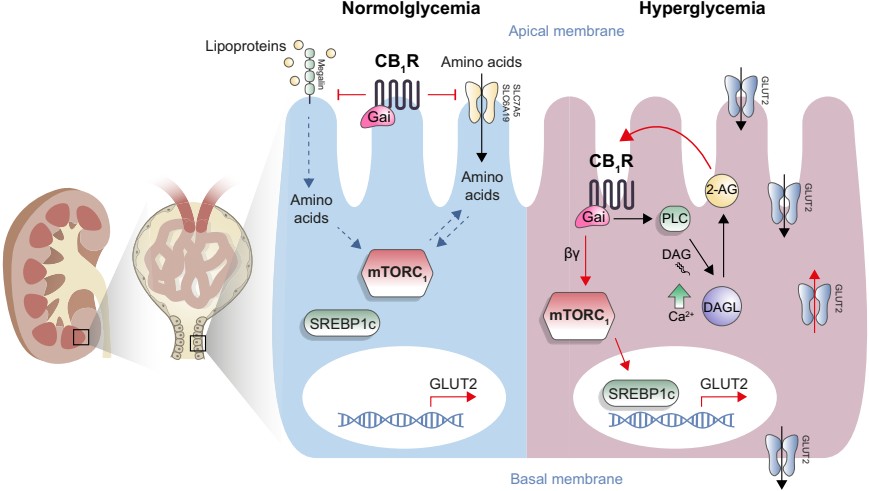

**Fig. 8 A proposed mechanism by which renal proximal tubule CB₁R regulates mTORC1 activity in health and disease.** In normoglycemia CB₁R is responsible to maintain mTORC1 normal activation by preventing excess of amino acids uptake (left). In hyperglycemia, CB₁R-mediated mTORC1 over-activation results in upregulating GLUT2 transcription, consequently enhancing glucose absorption and tubular damage (right).

the development of DKD in Akita-RPTC-GLUT2$^{-/-}$ mice. Proximal tubular GLUT2 is known to be upregulated and translocated to the brush border membrane of the RPTCs where it facilitates glucose reabsorption during diabetes[55,95], and may increase glucose-induced tubular damage and tubule-interstitial fibrosis. The contribution of GLUT2 recruitment to the brush border membrane of the RPTCs in facilitating increased glucose reabsorption in diabetes is not clear, since kidney glucose reabsorption is also mediated via SGLT2 and SGLT1[96]. Possibly, in diabetes, luminal glucose concentrations exceed a point by which SGLT's glucose uptake is in its full capacity, GLUT2 recruitment to the apical membrane enables glucose "leakiness" back to the lumen, facilitating apical glucose recycling to promote sodium reabsorption through SGLTs[6]. Moreover, rising blood glucose levels hamper the tubule-to-blood glucose gradient, which is essential for glucose flux through basolateral GLUT2[97].

Therefore, to maintain 'normal' tubular glucose levels, GLUT2 is recruited to the apical brush border membrane to assist in glucose transfer into the cell. Marks and colleagues found that STZ-induced diabetes in rats increases facilitative glucose transport at the brush border membrane by 67.5%, mainly due to GLUT2 translocation[55]. Our novel in vivo PET-MRI data emphasize the key role of GLUT2 in glucose reabsorption in diabetes, since its ablation in RPTCs normalized the diabetes-induced glucose uptake to the kidney. In addition, Umino et al. suggested that activation of the GLUT2/importin-α1/HNF-1α pathway by basolateral HG enhances SGLT2 expression in RPTCs; this was preventable by GLUT2 silencing/inhibition[98]. These findings are also supported by our data, demonstrating that genetic deletion of GLUT2 in RPTCs decreased SGLT2 expression, regardless the status of glycemia, suggesting that tight crosstalk between these two glucose transporters regulates kidney glucose handling and function.

Our results may also have translational aspects when one considers the therapeutic potential of pharmacologically targeting GLUT2 for the treatment of DKD. However, GLUT2 inhibition is unlikely to discriminate between apical and basal GLUT2 and might have major systemic side-effects as GLUT2 plays a role in glucose transport in other tissues, including the gut, islets, the liver, and neurons. Moreover, kidney glucose toxicity may remain an issue if glucose is taken up into RPTCs via SGLT2 but cannot exit to the interstitium and the blood at the basolateral side. Therefore, from this perspective, targeting the CB$_1$R and its downstream signaling molecules identified here may be a more fruitful therapeutic strategy. An alternative approach would be to target the recruitment machinery of GLUT2 to the brush border membrane to restrain the development of DKD rather than targeting the transporter itself. Further work is required to test which option might become a vital therapeutic approach for DKD.

## Methods

**Animals and the experimental protocol**. The Institutional Animal Care and Use Committee of the Hebrew University (AAALAC accreditation #1285; Ethic approval number MD-19-15784) approved the experimental protocol used. Animal studies are reported, in compliance with the ARRIVE guidelines[99]. The current experiment is based on the rule of the replacement, refinement, or reduction. All the animals used in this study were males under C57BL/6J background that were housed under specific pathogen-free (SPF) conditions, up to five per cage, in standard plastic cages with natural soft sawdust as bedding. The animals were maintained under controlled temperature of 22–24 °C, humidity at 55 ± 5%, and alternating 12 h light/dark cycles (lights were on between 7:00 and 19:00 h), and provided with chow food (Cat#, NIH-31 rodent diet) and water ad libitum. To elucidate the role of the proximal tubular CB$_1$R in the pathogenesis of DKD, we generated a novel diabetic mouse strain that lacks CB$_1$R specifically in the RPTCs (AK-RPTC-CB$_1$R$^{-/-}$) by crossing the Akita$^{Ins2+/C96Y}$ mouse model for type I diabetes with RPTC-CB$_1$R$^{fl/fl;sglt2Cre}$ [RPTC-CB$_1$R$^{-/-}$] characterized previously[21,25]. To generate diabetic mice lacking GLUT2 in RPTCs, we first crossed mice containing two *loxP* sites flanking the open reading frame of the GLUT2 gene (GLUT2$^{fl/fl}$; described in[100]) with the iL1-sglt2-Cre line[101]. Then RPTC-GLUT2$^{fl/fl;sglt2Cre}$ (RPTC-GLUT2$^{-/-}$) were crossed with Akita$^{Ins2+/C96Y}$ to generate AK-RPTC-GLUT2$^{-/-}$ mice. To generate diabetic mice lacking RPTOR in RPTCs, we first crossed mice containing two *loxP* sites flanking the open reading frame of the RPTOR gene (*Raptor*$^{fl/fl}$; #013188, Jackson Laboratories, Bar Harbor, ME) with the iL1-sglt2-Cre line[101]. Then, RPTC-RPTOR$^{fl/fl;sglt2Cre}$ (RPTC-RPTOR$^{-/-}$) were crossed with Akita$^{Ins2+/C96Y}$ to generate AK-RPTC-RPTOR$^{-/-}$ mice[58]. To generate mice lacking TSC in RPTCs (WT-RPTC-TSC$^{-/-}$), we crossed mice containing two *loxP* sites flanking the open reading frame of the TSC gene (TSC$^{fl/fl}$; Tsc2$^{tm2.1Djk}$/Mmjax, # 37154, Jackson Laboratories, Bar Harbor, ME) with the iL1-sglt2-Cre line[101]. To generate diabetic mice mutated for both CB$_1$R (homozygous) and TSC (heterozygous) in the RPTCs (AK-RPTC-CB$_1$R$^{-/-}$TSC$^{+/-}$), we crossed the AK-RPTC-CB$_1$R$^{-/-}$ mice with the WT-RPTC-TSC$^{-/-}$ mice.

At three weeks of age, littermates were divided into four groups according to their genotypes: WT-Cre$^-$, WT-Cre$^+$, AK-Cre$^-$, and AK-Cre$^+$. Cre$^-$ or Cre$^+$ refers to the presence or deletion of the gene of interest (*CB$_1$R, GLUT2, Raptor,* and *TSC*), respectively. All animals were homozygous for *flox*, except for TSC. The mice were monitored weekly for their blood glucose levels and body weight until they were sixteen weeks of age. Mice were subjected to a complete metabolic and kidney analysis. 24 h urine was collected one week before euthanasia using mouse

metabolic cages (CCS2000 Chiller System, Hatteras Instruments, NC, USA). At week 16, the mice were euthanized by a cervical dislocation under anesthesia, the kidneys and pancreases were removed and weighed, and samples were either snap-frozen or fixed in buffered 4% formalin. Trunk blood was collected for determining the biochemical parameters.

CB$_1$R activation in WT mice was done in 8-week-old, C57Bl/6J male mice. WIN 55,212-2 (3 mg/kg), or a vehicle (Veh; 1% Tween80, 4% DMSO in saline) was injected ip, and after 6 h the mice were euthanized by a cervical dislocation under anesthesia, the kidneys were removed and were either snap-frozen or fixed in buffered 4% formalin.

**Materials**. ACEA, Wortmannin, WIN 55,212-2, and Rapamycin were purchased from Cayman Chemical (USA). PTX was purchased from Sigma-Aldrich (Israel). YM-254890 was purchased from AdipoGen (USA). JD5037 and DO34 were purchased from MedChemExpress (China).

**Blood and urine biochemistry**. Serum insulin and urine albumin were measured by ELISAs (Millipore and Bethyl Laboratories, respectively). Serum urea, urine/serum glucose, and creatinine were determined using a Cobas C-111 chemistry analyzer (Roche, Switzerland). Blood urea nitrogen (BUN) was calculated by the serum urea levels (*BUN mg/dL = Urea mM × 2.801*). CCr was calculated using urine and serum creatinine levels (*CCr mL h$^{-1}$ = Urine creatinine mg/dL × Urine volume/Serum creatinine mg/dL × 24 h*). The urine levels of KIM-1 were measured by ELISA kit (R&D Systems, MN, USA).

**Histopathological Analyses**. Paraffin-embedded kidney and pancreas sections (5 μm) from each group (5–6 animals per group) were stained with hematoxylin and eosin or Periodic Acid–Schiff (PAS), followed by hematoxylin staining. In addition, kidney sections were stained for collagen types I and III deposition by using Masson's trichrome (Abcam, ab150686) staining according to the manufacturer's procedures. Kidney and pancreas images were captured with a Zeiss AxioCam ICc5 color camera mounted on a Zeiss Axio Scope.A1 light microscope, and taken from 10 random 40× fields of each animal. Mesangial expansion, glomerular, and Bowman's space cross-sectional areas were quantified using ZEN imaging software (Zeiss, Germany). Masson's trichrome positive areas were quantified using Image J software with a minimum of 10 random kidney images per mouse.

**Fluorescence immunohistochemistry**. Kidney sections were deparaffinized and hydrated. Heat-mediated antigen retrieval was performed with 10 mM citrate buffer pH 6.0 (Thermo Scientific, IL, USA). Unspecific antigens were blocked by incubating sections for 1 h with 5% goat serum (VE-S-1000, Vector Laboratories). The sections were stained with a Rabbit anti-LRP2 (ab76969, Abcam, 1:1000) antibody, followed by incubation with a Goat anti-Rabbit-AF488 antibody (ab150077, Abcam, 1:500). Sections were mounted with a mounting medium with DAPI (H-1200, Vector) and photographed using the LSM 700 imaging system (Zeiss). The relative fluorescent intensity (RFI) was measured using the ImageJ software (NIH, Bethesda, MD).

**Cell culture**. Human primary RPTCs (Lonza) were cultured in REGM BulletKit medium (Lonza), as described previously[25]. HEK-293 cells were cultured in HG-DMEM supplemented with 10% fetal bovine serum and 100 IU/mL penicillin/streptomycin (Biological industries, Israel) at 37 °C in a humid atmosphere with 5% CO$_2$.

To test the effect of CB$_1$R blockade and its signaling pathways, hRPTCs or HEK-293 cells were cultured overnight with serum-free medium (SFM) REGM or DMEM low glucose (LG), respectively, supplemented with 0.1% BSA. One hour before exposing the cells to HG (30 mM) or activating CB$_1$R with arachidonyl-2'-chloroethylamide (ACEA; 10 μM), the cells were pretreated with JD5037 (100 nM), DO34 (100 nM), Wortmannin (500 nM), Pertussis toxin (PTX; 100 ng/mL), YM254890 (1 μM), or Rapamycin (100 nM) in SFM. Gradual time frames were used to detect the effect of the treatment on S6 and AKT phosphorylation as well as for eCB measurements (1 h), evaluating GLUT2 transcriptional activity (3 h), and assessing GLUT2 protein expression (24 h).

**Immunofluorescence**. hRPTCs were seeded on eight chambered coverglass (In Vitro Scientific) in REGM. The cells were starved overnight in SFM-REGM and pretreated for 1 h with JD5037 (100 nM) or rapamycin (100 nM) and then for an additional 24 h with HG (30 mM). Next, the cells were fixated with 4% PFA and permeabilized with 0.25% Triton x100. After an additional two hours of blocking in 2% BSA, cells were incubated overnight with Rabbit anti-GLUT2 (#AGT022, Alomone, 1:500) or Rabbit anti-SREBP1 (ab193318, Abcam, 1:500). Then, cells were incubated with Donkey anti-Rabbit-APC (#711-136-152, Jackson, 1:1000) or Goat anti-Rabbit-AF488 (ab150077, Abcam, 1:1000) secondary antibodies respectively, for 1 h and Hoechst for 15 minutes, and next photographed using a IX-73 fluorescent microscope (OLYMPUS) or a Confocal A1R microscope (Nikon). RFI was measured using Image J software.

**Vectors**. The hGLUT2 promoter 1043 bp site with the XohI/HindIII restriction sites were amplified via PCR from hRPTC DNA and cloned into the multiple cloning region of the pGL3 luciferase reporter vector (Promega, USA), generating the pGL3-GLUT2 vector. The *Renilla* luciferase vector (Promega, USA) was used as a co-reporter vector and for expression control. The CB₁R-TK-d64 plasmid contained a short version of the CB₁R gene under the thymidine kinase (TK) promoter, enabling CB₁R to translocate to the plasma membrane more easily than the full-length CB₁R[64] (Supplementary Fig. 6).

**GLUT2 transcription luciferase reporter assay**. HEK-293 cells were transiently transfected using the Lipofectamine™ 3000 transfection reagent (Thermo Fisher Scientific, USA) with the pGL3-GLUT2 and *Renilla* vectors with/without the CB₁R-TK-d64 plasmid[64]. One day following the transfection, cells were starved overnight in SFM and treated for 3 h with HG (30 mM) or CB₁R agonist-ACEA (10 μM), with/without 1 h pretreatment with JD5037 (100 nM) or Rapamycin (100 nM). The Dual-Glo Luciferase Assay System (Promega, USA) was used to detect GLUT2 promoter transcriptional activity. The Relative Response Ratio (RRR) was calculated for each treatment and compared to the control group.

**Promoter-binding transcription-factor profiling assay**. A list of ~120 TFs that can bind to the 1043 bp hGLUT2 promoter site was prepared using TFBIND software. A modified list of 47 relevant TFs was screened for their ability to bind to the hGLUT2 promoter using a promoter-binding TF profiling array (Signosis, USA). An assay was performed according to the manufacturer's instructions. Briefly, nuclear extract (NE) was isolated from $1 \times 10^7$ hRPTCs, and treated/untreated with HG for 3 h, by using a nuclear protein extraction kit (Signosis, USA). The reaction mixture was prepared using 15 μL of the TF binding buffer, 3 μL of the probe, 10 μg of nuclear extract, and 5 μL of the hRPTC GLUT2 promoter fragment (1043 bp), and incubated at room temperature for 30 min to allow for the formation of the TF-DNA complex. Unbound probes were separated from the complex, whereas bound probes were eluted and then hybridized to the plate and incubated overnight at 42 °C. Bound probes were detected using an HRP-streptavidin conjugate incubated with the chemiluminescent substrate. Luminescence is reported as relative light units (RLUs) on the Multi-Mode Microplate Reader SpectraMax iD3 (Molecular Devices, USA).

**Small interfering RNA treatment**. Small interfering RNA (siRNA) transfection against SREBP1 (sc-36557, Santa Cruz) was performed in hRPTCs using the siRNA reagent system (sc-45064, Santa Cruz) according to the manufacturer's instructions.

**In vivo micro PET-MRI scanning**. Experiments were performed at the Wohl Institute for Translational Medicine at Hadassah Hebrew University Medical Center. PET-MRI images were acquired on a 7T 24 cm bore, a cryogen-free MR scanner based on the proprietary dry magnet technology (MR Solutions, Guildford, UK) with a 3-ring PET insert that uses the latest silicon photomultiplier (SiPM) technology[102]. The PET subsystem contains 24 detector heads arranged in three octagons of 116 mm in diameter. For MRI acquisition, a mouse quadrature RF volume coil was used. Mice were anesthetized with isoflurane vaporized with O₂. Isoflurane was used at 3.0% for induction and at 1.0–2.0% for maintenance. The mice were positioned on a heated bed, which allowed for continuous anesthesia and breathing rate monitoring. To determine the distribution of [¹⁸F]-FDG in mice, the tracer was injected into the tail vein (230 ± 30 mCi in 200 mL). Mice were subjected to 31 min dynamic PET scans; a homemade small catheter was inserted into the proximal tail vein and the tracer was injected after positioning the mouse in the micro-PET/MRI scanner. For dynamic scans, the acquired data were binned into 25 image frames (1 × 60, 6 × 10, 8 × 30, 5 × 60, and 4 × 300 s). During the PET, scans were acquired, T1 & T2; weighted coronal spin echo images were collected for anatomical evaluation. Coronal T1 weighted images were acquired using the following parameters: TR = 1100 ms, TE = 11 ms, echo spacing = 11 ms, FOV = 6 × 3 cm, slice thickness = 1 mm, 4 averages. Coronal T2-weighted images were acquired using the following parameters: TR = 4000 ms, TE = 45 ms, echo spacing = 15 ms, FOV = 6 × 3 cm, slice thickness = 1 mm, and 4 averages. Images were analyzed using VivoQuant pre-clinical image post-processing software (Invicro). PET-MRI raw data were processed using the standard software provided by the manufacturers. PET data were acquired in list-mode, histogrammed by Fourier re-binning, and reconstructed using the 3D-OSEM algorithm, with standard corrections for random coincidences, system response, and physical decay applied. The reconstructed PET images from the PET/MR scanner were quantitated using a measured system-specific ¹⁸F calibration factor to convert reconstructed count rates per voxel to activity concentrations (%ID/g). Manual tissue segmentation of kidneys, liver, muscle, inferior vena cava (IVC), and bladder was carried out on co-registered 3D MR images. The regions of interest were then used to calculate tissue radiotracer uptake from the reconstructed PET images.

**Western blotting**. Kidney or cell homogenates were prepared in a RIPA buffer (25 mM Tris-HCl pH 7.6, 150 mM NaCl, 1% NP-40, 1% sodium deoxycholate, 0.1% SDS). Kidney homogenates were prepared by using the BulletBlender® and zirconium oxide beads (Next Advanced, Inc., NY, USA). Protein concentrations

were measured with the Pierce™ BCA Protein Assay Kit (Thermo Scientific, IL, USA). Samples were resolved by SDS-PAGE (4–15% acrylamide, 150 V) and transferred to PVDF membranes using the Trans-Blot® Turbo™ Transfer System (Bio-Rad, CA). Membranes were then incubated for 1 h in 5% milk (in 1× TBS-T) to block unspecific binding. Membranes were incubated overnight with Rabbit/Mouse anti-GLUT2 (#AGT022, Alomone, 1:1000; #720238, Invitrogen, 1:1000), Rabbit anti-phosphorylated-S6 ribosomal protein (#5364, Cell Signaling, 1:50000), Rabbit anti-phosphorylated-AKT (#9272, Cell Signaling, 1:500), Rabbit/Mouse anti-SREBP1 (#ab193318, #ab3259, Abcam, 1:500), Rabbit anti-SLC6A19 (#ab180516, Abcam, 1:500), Rabbit anti-SLC7A5 (#5347 S, Cell Signaling, 1:500), Rabbit anti-SGLT2 (#ab85626, Abcam) Goat anti-DAGLα (#ab81984, Abcam, 1:500), Rabbit anti-DAGLβ (#ab191159, Abcam, 1:1000), Rabbit anti-LRP2 (ab76969, Abcam, 1:500), and Rabbit anti-CB₁R (#301214, Immunogen, 1:500) antibodies at 4 °C. Anti-Rabbit/Mouse horseradish peroxidase (HRP)-conjugated secondary antibodies (#ab97085, #ab98799, Abcam, 1:2500) were used for 1 h at room temperature, followed by chemiluminescence detection using Clarity™ Western ECL Blotting Substrate (Bio-Rad, CA), blot imaging was done using ChemiDoc™ Touch Imaging System (Bio-Rad, CA). Densitometry was quantified using Bio-Rad CFX Manager software. Quantification was normalized to Mouse anti-β actin antibody (#ab49900, Abcam, 1:30000) or Rabbit anti-VCP (#ab204290, Abcam, 1:1000). Phosphorylated S6 and AKT were normalized to total Rabbit anti-S6 ribosomal protein (#2217, Cell Signaling, 1:500) and Rabbit anti-total AKT (#4058, Cell Signaling, 1:500), respectively. Nuclear proteins were normalized to Mouse anti-Fibrillarin (#ab4566, Abcam, 1:500).

**Real-time PCR**. Total kidney or cell mRNA was extracted using Bio-Tri RNA lysis buffer (Bio-Lab, Israel), followed by DNase I treatment (Thermo Scientific, IL, USA), and reverse transcribed using the Iscript cDNA kit (Bio-Rad, CA). Real-time PCR was performed using iTaq Universal SYBR Green Supermix (Bio-Rad, CA) and the CFX connect ST system (Bio-Rad, CA). The primers used to detect mouse or human genes are listed in Supplementary Tables 1, 2. Mouse and human genes were normalized to *Ubc* or RPLP, respectively.

**Sample preparation and endocannabinoid measurements by LC-MS/MS**. The eCBs and related lipid measurements were performed in two independently laboratories. eCBs were extracted, purified, and quantified from RPTC lysates. In brief, A. (Tam Laboratory, Hebrew University) hRPTCs were scraped from the culture plates in ice-cold Tris Buffer, homogenized using sonication, and protein concentration was determined. Samples were then supplemented with an ice-cold Extraction buffer (1:1 Methanol/Tris Buffer+ Internal Standard) and Chloroform/Methanol (2:1), vortexed and centrifuged. The lower organic phase was transferred into borosilicate tubes; this step was repeated three times by adding ice-cold Chloroform to the samples and transferring the lower organic phase into the same borosilicate tubes. The samples were then dried and kept overnight in −80 °C, and then reconstitute with ice-cold Chloroform and Acetone then kept in −20 °C for 30 min and centrifuged to precipitate proteins. The supernatant was then dried and reconstituted in ice-cold LC/MS grade Methanol. LC-MS/MS was analyzed on an AB Sciex (Framingham, MA, USA) QTRAP® 6500 + mass spectrometer coupled with a Shimadzu (Kyoto, Japan) UHPLC System. Liquid chromatographic separation was obtained using 5 μL injections of samples onto a Kinetex 2.6 μm C18 (100*2.1 mm) column from Phenomenex (Torrance, CA, USA). The autosampler was set at 4 °C and the column was maintained at 40 °C during the entire analysis. Gradient elution mobile phases consisted of 0.1% formic acid in water (phase A) and 0.1% formic acid in acetonitrile (phase B). eCBs were detected in a positive ion mode using electron spray ionization (ESI) and the multiple reaction monitoring (MRM) mode of acquisition, using d₄-AEA as internal standard (IS). The collision energy (CE), declustering potential (DP), and collision cell exit potential (CXP) for the monitored transitions are given in Supplementary Table 5. The levels of AEA, 2-AG, OEA, PEA, and AA in samples were measured against standard curves, and normalized to the RPTC lysate protein concentration.

B. (Gertsch Laboratory, University of Bern) similar to Schuele et al.[103], the scraped hRPTCs were homogenized in 0.1 M FA by three times freeze-thaw cycles and sonication. The protein concentration was determined with an aliquot of the sample using BCA. The remaining homogenate was extracted in 9:1 ethyl acetate:hexane 0.1% FA solution (3:1 ratio organic to water part). After a drying the organic layer, the sample was reconstituted in 35 μL 80% acetonitrile and 10 μL was injected per measurement into the system. LC-MS/MS was conducted on an AB Sciex (Framingham, MA, USA) Triple Quad™ 5500 mass spectrometer coupled to a Shimadzu (Kyoto, Japan) UHPLC System. For the chromatographic separation, a C18 column (3 μm particle size; 2 × 50 mm, Dr. A. Maisch HPLC GmbH, Ammerbuch-Entringen, Germany) was used. In a positive mode, the mobile phase A was water containing 2 mM NH₄Ac 0.1% FA, and mobile phase B contained methanol and 2 mM NH₄Ac. In a negative mode, mobile phase A consisted of water 2 mM NH₄Ac 0.1% FA, and mobile phase B was acetonitrile 0.1% FA. The flow rate was set to 0.3 mL/min. eCBs were detected in a positive ion mode (for SAG, 2-AG, and ethanolamines) and AA in a negative mode using ESI and the MRM mode. The molecular ions and fragments for each compound are presented in Supplementary Table 5. Data acquisition and analysis were performed using Analyst software 1.6. For all analyses, a linear regression without weighing was applied. The levels of AEA, 2-AG, SAG, OEA, PEA, and AA in samples were

measured against standard curves, and normalized to the RPTC lysate protein concentration.

**Sample preparation and amino acid measurement by LC-MS/MS.** Amino acids were extracted, purified, and quantified from kidney lysates or from cultured hRPTCs. In brief, kidney samples were weighed, added with ice-cold methanol, sonicated to lysates, and centrifuged. The supernatant was diluted 1:10 in methanol with an internal standard (S-(2-Aminoethyl)-L-cysteine hydrochloride). hRPTCs were starved overnight in SFM DMEM F-12, w/o amino acids (USBiological, USA) and treated with/without JD5037 (100 nM) for 1 h. Then 0.05447 mg/mL L-Isoleucine, 0.05905 mg/mL L-Leucine, and 0.05285 mg/mL L-Valine were added to the medium and treated for 3 h; then the cells were lysed using a solvent containing methanol, acetonitrile, and water at a ratio of 5:3:2, respectively, with 5 μM of the internal standard. LC-MS/MS analyses were conducted under reverse phase conditions on a Sciex (Framingham, MA, USA) QTRAP® 6500+ mass spectrometer coupled with a Shimadzu (Kyoto, Japan) UHPLC System. Liquid chromatographic separation was achieved using 5 μL injections of samples onto an Intrada Amino Acids column 3 μm (150*2 mm) from Imtakt Corp. (Kyoto, Japan). The autosampler was set at 10 °C and the column was maintained at 40 °C during the entire analysis. Gradient elution mobile phases consisted of 100 mM ammonium formate in water (phase A) and 0.1% formic acid in acetonitrile (phase B). Gradient elution (400 μL/min) was held at 14% A for the first 3.75 min, followed by a linear increase towards 55% A in 5.5 min, a linear increase towards 100% A in 1 min, and held at 100% A for 6 min. Amino acids were detected in a positive ion mode using ESI and the Advanced Scheduled MRM mode of acquisition. The Turbo Spray IonDriveTM Turbo V source temperature was set at 650 °C, with the ion spray voltage at 5000 V. The curtain gas was set at 30.0 psi. The nebulizer gas (Gas 1) was set to 50 psi, and the turbo heater gas (Gas 2) was set to 60 psi. The CE, DP, and the CXP for the monitored transitions are presented in Supplementary Table 6. Data acquisition was performed on a Dell Optiplex XE2 computer using Analyst 1.7.1 and data was analyzed using Sciex OS Software. The levels of the amino acids in samples were measured against standard curves and normalized to the kidney weight.

**Statistics.** Values are expressed as the mean ± SEM. Unpaired Two-tailed Student's t-test was used to determine the differences between two groups. Results in multiple groups were compared by one-way ANOVA followed by one-sided Tukey test, using GraphPad Prism v6 for Windows (San Diego, CA). Significance was set at $P < 0.05$.

**Reporting summary.** Further information on research design is available in the Nature Research Reporting Summary linked to this article.

## Data availability

All data that support the findings of this study are available within the article, its Supplementary Information or Source Data files. Primers lists are provided in Supplementary Tables 1, 2 (in the Supplementary Information file). Uncropped gels are available in the Supplementary Information file. Source data are provided with this paper.

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

## Acknowledgements

We would like to thank Prof. Boaz Tirosh for his critical advice on mTORC1 signaling, and Dr. Dinorah Barasch for her technical assistance in LC-MS/MS analysis. This work was supported by an ERC-2015-StG grant (#676841), an Israel Science Foundation (ISF) grant (#158/18), and a JDRF grant (1-INO-2022-1128-A-N) to J.T. The work of G.S. was supported by the National Research, Development and Innovation Office grant NKFI-6/FK_124038.

## Author contributions

L.H., M.A., and S.H. conducted the experiments and analyzed the data. A.N., S.G., and J.G. conducted the LC-MS/MS analyses. A.K.L. and G.L. assisted with the animal experiments. G.S. provided experimental reagents. B.T. provided the GLUT2$^{fl/fl}$ mice. R.A. performed the PET-MRI studies. L.H. and J.T. designed and supervised the experiments and wrote the manuscript. All co-authors contributed to writing the manuscript.

## Competing interests

The authors declare no competing interests.
