## [Peer Review File · Nature Communications]

Opposite physiological and pathological mTORC1-mediated roles of the CB1 receptor in regulating renal tubular functionReviewers' comments:

Reviewer #1 (Remarks to the Author):

Liad H et al. examined the alteration of CB1 receptor-mTORC1 axis in relation to GLUT2 expression in the development of diabetic kidney disease in type 1 diabetic model and cultured proximal tubular cells exposed to high glucose. The authors provided the molecular mechanism of CB1 receptor-mTORC1 axis activation results in SREBP1-induced GLUT2 overexpression, thereby developing diabetic kidney disease phenotype. In contrast, CB1 receptor might restrict mTORC1 overactivation by inhibiting nutrient reabsorption in normal condition.

Comments;

- 1) Abstract, introduction, and in the text; diabetic CKD and or CKD should be changed to diabetic kidney disease (DKD)
- 2) Creatinine clearance did not provide the true GFR and the authors must measure inulin clearance.
- 3) I am wondering whether RPTCs specific CB1R knockout by which mechanisms affects proteinuria, albuminuria, glomerular hypertrophy, mesangial expansion. Have you performed the study regarding to glomerular podocytes?
- 4) The authors have published previously the GLUT2 translocation in Ref 23. However, the reviewer cannot understand whether GLUT2 is located at apical site of proximal tubular cells. Have you confirmed the expression of SGLT2 or 1 in your study? How did GLUT2 downregulation in CB1R knockout in RTPCs cause glucosuria?
- 5) Regarding the mechanisms by which CB1R activate GLUT1 expression and mTORC1 activation in hRPTCs exposed to high glucose, these data have to confirmed in vivo.
- 6) I am also wondering whether GLUT knockout in RTPCs mainly induced DKD. How do you discriminate the contribution of SGLT2 and/or SGLT1 in RTPCs?
- 7) Discussion section is too long and must shorten based on your data.

Reviewer #2 (Remarks to the Author):

Hinden et al. report that CB1R upregulates GLUT2 via mTORC1 and SREBP1, specifically in the setting of hyperglycemia. This work follows on a recently published paper in which the authors established that diabetic-induced upregulation of renal GLUT2 is mitigated by peripheral pharmacological blockade or genetic deletion of CB1R in RTPCs, thereby preventing CKD.

The effects of deletion of CB1R on phospho-S6 and Glut2 in the Akita model are quite clear (Figure 2A, 2C). However, in contrast to the proposed model, there is very little impact of Raptor KO on Glut2 levels in vivo (Figure 2J) and little impact of hyperglycemia on Glut2 in vitro (Fig 2N). The experiments in Figure 3 show essentially no effect of activation of CB1R on phospho-S6 (densitometry shows a ~30% increase in 3A, but this is extremely subtle by eye). The conclusions related to SREBP1 are not strongly supported by the data in Figure 4, which include a nuclear/cytoplasmic fractionation experiment in which key controls are missing (Fig 4O). Overall, there are issues with data quality (particularly immunoblotting) throughout. Therefore, while the in vivo experiments in Figure 1 and 6 are solid and confirm prior work by this group, the mechanistic experiments do not support the authors' key conclusions, as detailed below. It appears that the authors have frequently over-interpreted their data, and/or relied on densitometry (often of overexposed blots and/or blots with technical artifacts) to support conclusions that are not visually evident.

Major concerns:

Fig 1. The "n" in the figure legends does not seem to correspond to the number of dots in the dot blots. This is evident in Figure 1c,d,e, and in other figures throughout the manuscript

Fig 2. As noted above, Figure 2J clearly reveals that mTORC1 has little effect on Glut 2. The authors show densitometry results (again, the number of dot on the dot blot does not seem to

match the number of lanes on the gel) but parts of this gel are obscured by artifact. Regardless, given the results in 2A and 2C, this result is pretty conclusive that Glut2 is NOT mTORC1 dependent.

Fig 2 and elsewhere. The authors should be cautious with their use of densitometry on overexposed immunoblots or blots with artifacts (For example Figure 2C, 2J, and many others). In Figure 2B, both the phospho Akt and the total Akt are overexposed on the left panels, and there appears to be no phospho-Akt on the right; the densitometry in 2E cannot be trusted; there are many other examples including 2N.

Fig 2H and 2I. It is not clear how this correlation has been quantified, what do the dots represent?

Figure 2 R, S, T. It is unclear what these panels represent; eCB tone is not explained.

Fig.3 The effects of JD5037 seem to be minimal, if any, in contrast to the conclusions of the authors. Densitometry should not be used for blots such as 3D.

Figure 3J appears to have a transfer artifact in the Western, affecting the phospho-S6 lanes.

Figure 4O lacks controls for the cytoplasmic fraction. The levels of fibrillarlin differ dramatically – was this a control for the nuclear fraction? It is not mentioned in the text or legend. If this is a control, it brings into question the entire experiment since it is perhaps 100X lower in the control cells vs. the others. There is no clear difference between the relative localization of SREBP1 (cytoplasmic vs. nuclear) in any of the conditions. Was the densitometry done on multiple biologic replicates?

Figure 6 – if the mechanism is truly as proposed in Figure 7, the authors should treat the mice with Rapamycin or do a Raptor KO, and this should prevent CKD.

Reviewer #3 (Remarks to the Author):

This is a very elegant study in which the authors have described a typical example of how the endocannabinoid system, particularly when acting via CB1 receptors, can play both physiological and pathological roles, as it is often the case with endogenous pro-homeostatic systems that can get deranged (and become maladaptive instead) during prolonged perturbations of homeostasis. The authors have elegantly dissected, in renal proximal tubular cells (RPTCs), almost all the molecular events going from CB1 activation or overactivation and the final cellular response, which under physiological conditions, is concerned with regulation of amino acid uptake, and under conditions of hyperglycemia, with upregulation of glucose uptake. In the former case, the effect of CB1 is due to down-regulation of mTORC1 overactivity, whereas in the latter case to its overactivation. I have suggestions for only text corrections, nevertheless important to ameliorate the text:

- 1) Starting from the title, I had some occasional problems in following the syntax of the text. The title as such is not clear, and I suggest something like "Opposite physiological and pathological mTORC1-mediated roles of CB1 receptors in the control of renal tubular function". Likewise, sentences on on page 4, lines 8-10, lines 13-14 and line 20, should be rephrased to afford clearer messages. On page 5, the long conclusive section of the Introduction is inappropriate for this part of the article, as it described the results of the study. It should be moved in the Results or Discussion, since the Introduction should only be used to provide the background and rationale for the study, and indicate the general experimental approach used to achieve the aims.
- 2) Not everybody knows what Akita mice are, and these guys are never mentioned in the Introduction. Therefore, I found the first mention of these mice on page 6, line 7, quite confusing.
- 3) On page 7, the authors mention the results with measures of the levels of 2-AG biosynthetic precursor, 2-AG and 2-AG product, AA. I agree that the observed changes fit with the finding of elevated 2-AG biosynthesis, but also with its elevated degradation. This should be mentioned in the Discussion. If the authors also mentioned the expression of DAGLalpha and beta in RPTCs, and found an increase of either or both enzymes following hyperglycemia, this would also fit nicely in

the results. Indeed, Sampaio et al., 2015 (which should be quoted in this section of the Results too) did find that a model of RPTCs express more DAGLbeta than alpha, and it would be nice to see which of the two enzymes was responsible for 2-AG biosynthesis in hRPTCs;

4) The Discussion is probably a bit too long, and yet the authors have not mentioned that it is not the first time that hyperglycemia leads to elevated levels of endocannabinoids in vitro, as shown for the first time by Matias et al. *J Clin Endocrinol Metab* 2006 in beta-cells. These authors also showed for the first time that diabetic hyperglycemic patients exhibit elevated AEA and 2-AG levels in the plasma, and therefore this paper and these previous findings should be mentioned;

5) On page 20, line 16, the authors mention the possibility that CB1 may regulate Na⁺/K⁺-ATPase activity in RPTCs; indeed, this has been shown to be the case by Sampaio et al., *Biochem Pharmacol* 2018, and therefore also this paper should be mentioned.

6) As to the possible mechanism by which CB1 might lead to elevated 2-AG levels, this has been shown to occur via Gq-mediated elevation of intracellular calcium by Hegyi et al *Sci Rep* 2018, in astrocytes, whereas CB1 activation of the master regulator of 2-AG biosynthesis, phospholipase Cbeta, has been shown to occur again via Gq, in HEK-293 cells by Hermann et al, *Cell Mol Life Sci* 2003. These two papers should be quoted.

Reviewer #1 (Remarks to the Author):

Liad H et al. examined the alteration of CB1 receptor-mTORC1axis in relation to GLUT2 expression in the development of diabetic kidney disease in type 1 diabetic model and cultured proximal tubular cells exposed to high glucose. The authors provided the molecular mechanism of CB1 receptor-mTORC1axis activation results in SREBP1-induced GLUT2 overexpression, thereby developing diabetic kidney disease phenotype. In contrast, CB1 receptor might restrict mTORC1 overactivation by inhibiting nutrient reabsorption in normal condition.

Comments;

1) Abstract, introduction, and in the text; diabetic CKD and or CKD should be changed to diabetic kidney disease (DKD)

Amended throughout the text.

2) Creatinine clearance did not provide the true GFR and the authors must measure inulin clearance.

We and others [3-8] have already shown that CB₁R antagonism/absence ameliorates DKD; the aim of this paper is to add novel mechanistic information on how this effect is specifically mediated. Creatinine clearance is only one parameter used here to assess the kidney function of our novel diabetic KO mice. Although we agree that inulin clearance in mice is considered the most appropriate methodology to test kidney function, we weren't able to assess it due to the high number of null mice needed to undergo this procedure, in addition to all other tests. We believe

that our measurements of proteinuria, albuminuria, ACR, urinary KIM-1 levels, histopathology, and gene expression provide a clear picture of kidney function.

3) *I am wondering whether RPTCs specific CB1R knockout by which mechanisms affects proteinuria, albuminuria, glomerular hypertrophy, mesangial expansion. Have you performed the study regarding to glomerular podocytes?*

In our previous publication [9], we specifically described the mechanism by which RPTC-CB1R antagonism/absence reduces GLUT2 translocation from the basolateral to the apical/brush border membrane of the RPTCs, which in turn, reduces glucotoxicity, renal dysfunction, inflammation, and fibrosis. Since GLUT2 is specifically located in the RPTCs, and there is growing evidence that tubulopathy is a major contributor to DKD, our interests were primarily focused on these cells, and not on glomerular podocytes. Nevertheless, and as the reviewer is probably aware of, the role of glomerular podocyte CB1R in DKD has been extensively characterized, partially by us [3, 6, 8]. These studies were cited multiple times in the manuscript.

4) *The authors have published previously the GLUT2 translocation in Ref 23. However, the reviewer cannot understand whether GLUT2 is located at apical site of proximal tubular cells.*

The translocation of GLUT2 from the basolateral membrane to the apical/brush border membrane of the RPTCs under hyperglycemic conditions has been previously reported [10, 11]. Moreover, we focused on the same issue when we initially started evaluating the role of GLUT2 in RPTCs, and addressed this point specifically in our previous publication, demonstrating both *in vitro* and *in vivo* that high glucose conditions (as well as CB1R activation) promote GLUT2 translocation to the apical/brush border membrane of the RPTCs, an effect that is regulated by CB1R [9].

Have you confirmed the expression of SGLT2 or 1 in your study? How did GLUT2 downregulation in CB1R knockout in RPTCs cause glucosuria?

Although this specific issue was previously addressed by us [9], where we showed that CB1R blockade/deletion in RPTCs does not directly affect the expression of SGLT2 (see Supplementary Figure 11 in ref [7]), we decided to revisit the same issue in our current manuscript, and try to explain the increased glucosuria in RPTC-GLUT2 null mice. As can be seen, in the current manuscript the ablation of GLUT2 in RPTCs results in glucosuria and protects against renal injury (see Figure 7). These effects were associated with reduced gene and protein expression of SGLT2 under both normo- and hyper-glycemic conditions (see Figure S12). Therefore, the reduction in SGLT2 may also contribute to the increased glucose excretion into the urine and to the renal protection seen in RPTC-GLUT2 knockout mice. Although Vallon and colleagues [12] suggested that knocking out SGLT2 does not protect against DKD, our novel findings suggest that the absence of GLUT2, rather than SGLT2, is most likely responsible for the preserved kidney function.

Moreover, we further confirm these findings by specifically assessing glucose uptake in RPTC-GLUT2 knockout mice by using *in vivo* PET-MRI experiments, demonstrating increased renal glucose uptake in the diabetic mice, which was completely normalized in diabetic RPTC-GLUT2 knockout animals (see the new data in Figure 7w-x and Figure S11).

5) *Regarding the mechanisms by which CB1R activate GLUT2 expression and mTORC1 activation in hRPTCs exposed to high glucose, these data have to be confirmed in vivo.*

Our *in vivo* data clearly show that kidney CB1R expression is upregulated in Akita diabetic mice (Figure S1c-e) and that its deletion, specifically in RPTCs, reduces GLUT2 expression via

inhibiting mTORC1 signaling (Figure 2a-i). To further address this comment, we tested the effect of acute CB1R activation *in vivo* (by using WIN-55,212 in WT mice), and found that it resulted in activation of kidney mTORC1 signaling and elevated GLUT2 expression levels (see the new data in Figure S2a-c).

6) *I am also wondering whether GLUT knockout in RPTCs mainly induced DKD. How do you discriminate the contribution of SGLT2 and/or SGLT1 in RPTCs?*

Deletion of GLUT2 in RPTCs ameliorates DKD and does not induce it (see Figure 7). As mentioned in our response to your comment #4, we demonstrated that GLUT2 KO in RPTCs reduces SGLT2 expression (both at the gene and protein levels) regardless of their diabetic condition (see the new data in Figure S12). We suggest that nullification of GLUT2 in RPTCs may affect SGLT2 expression and consequently, it changes glucose reabsorption in the kidney (see the new data in Figure 7). Further work will be carried out in the future to determine whether GLUT2 and SGLT2 are directly linked.

7) *Discussion section is too long and must shorten based on your data. The discussion was shortened.*

Reviewer #2 (Remarks to the Author):

Hinden et al. report that CB1R upregulates GLUT2 via mTORC1 and SREBP1, specifically in the setting of hyperglycemia. This work follows on a recently published paper in which the authors established that diabetic-induced upregulation of renal GLUT2 is mitigated by peripheral pharmacological blockade or genetic deletion of CB1R in RPTCs, thereby preventing CKD.

The effects of deletion of CB1R on phospho-S6 and Glut2 in the Akita model are quite clear (Figure 2A, 2C). However, in contrast to the proposed model, there is very little impact of Raptor KO on Glut2 levels in vivo (Figure 2J) and little impact of hyperglycemia on Glut2 in vitro (Fig 2N). The experiments in Figure 3 show essentially no effect of activation of CB1R on phospho-S6 (densitometry shows a ~30% increase in 3A, but this is extremely subtle by eye). The conclusions related to SREBP1 are not strongly supported by the data in Figure 4, which include a nuclear/cytoplasmic fractionation experiment in which key controls are missing (Fig 4O). Overall, there are issues with data quality (particularly immunoblotting) throughout.

Therefore, while the in vivo experiments in Figure 1 and 6 are solid and confirm prior work by this group, the mechanistic experiments do not support the authors' key conclusions, as detailed below. It appears that the authors have frequently over-interpreted their data, and/or relied on densitometry (often of overexposed blots and/or blots with technical artifacts) to support conclusions that are not visually evident.

Although we thank the reviewer for highlighting the fact that our *in vivo* work is solid, we disagree with his comments regarding the *in vitro* and/or densitometry analyses. We would like to emphasize that only representative blots were shown, and that the densitometry analyses reflect an average of 3-5 independent experiments (*in vitro* biological replicates), and therefore, they are biologically relevant. Naturally, none of the immunoblots were manipulated or amended in any way. The main concern of this Reviewer was the seeming overexposure of our blots. In this

respect, we wish to emphasize that for the development of the blots, we used a CCD camera-based device, which clearly indicates if/when the exposure is saturated. This was not an issue in any of the blots. Moreover, modern devices measure HRP-luminescence at 12-bit depth or higher, whereas computer screens and printers struggle to visualize such bit-depth. That can lead to an apparent “overexposure” when watching the blots on a screen or printed paper; however, this does not affect the densitometry results *per se*. Gel artifacts are indeed present in some of the blots but, importantly, they rarely overlap with the bands so they do not raise major concerns. Finally, we wish to share with the Editor/Reviewers the raw images of the entire blots as primary data (see Appendix 1; Hinden et al_WB).

Major concerns:

Fig 1. The “n” in the figure legends does not seem to correspond to the number of dots in the dot blots. This is evident in Figure 1c,d,e, and in other figures throughout the manuscript

We agree with the reviewer and apologize for overlooking this issue. All the numbers were amended in the figure legends.

Fig 2. As noted above, Figure 2J clearly reveals that mTORC1 has little effect on Glut 2. The authors show densitometry results (again, the number of dot on the dot blot does not seem to match the number of lanes on the gel) but parts of this gel are obscured by artifact. Regardless, given the results in 2A and 2C, this result is conclusive that Glut2 is NOT mTORC1 dependent.

We completely disagree with the reviewer's interpretation of our data, since (i) deletion of RPTOR in RPTCs significantly reduces GLUT2 expression (see Figure 2j-m), and (ii) rapamycin, an mTORC1 inhibitor, ameliorates the hyperglycemia-induced GLUT2 transcription (see Figure 4e) as well as the nuclear localization of SREBP1 (see the new data in Figure 4s-v), which regulates GLUT2. To further support this conclusion and to strengthen our findings, we tested the effect of overactivation of mTORC1 on GLUT2 expression by using novel RPTC-TSC1 knockout mice that we developed, and found that mTORC1 overactivation specifically in RPTCs upregulates both GLUT2 and SREBP1 expression (see the new data in Figures S2d-g). Taken together, these findings DO SUGGEST that GLUT2 is indeed regulated by mTORC1.

Fig 2 and elsewhere. The authors should be cautious with their use of densitometry on overexposed immunoblots or blots with artifacts (For example Figure 2C, 2J, and many others). In Figure 2B, both the phospho Akt and the total Akt are overexposed on the left panels, and there appears to be no phospho-Akt on the right; the densitometry in 2E cannot be trusted; there are many other examples including 2N.

See our response to the general comments raised by this reviewer.

Fig 2H and 2I. It is not clear how this correlation has been quantified, what do the dots represent?

Each dot represents one mouse; the higher pS6 or pAKT is expressed within a given mouse the higher GLUT2 is expressed.

Figure 2R, S, T. It is unclear what these panels represent; eCB tone is not explained.

eCB 'tone' represents the profile of eCBs measured by LC-MS/MS in a given sample. Figure 2r-t and Figure S4 show the profile of different eCBs in cultured hRPTCs exposed to glucotoxic conditions (30 mM glucose) in the presence or absence of JD5037 (100 nM; a CB₁R antagonist). Our findings reveal that the 2-AG levels are elevated in high-glucose-treated cells, most likely due to increased activity, and not expression of its synthesizing enzyme, DAGL, and that this effect is CB₁R-dependent, since JD5037 normalizes it. Following a comment by Reviewer #3, we also expanded the data set to include DAGL expression (see the new data in Figure S5). The text was amended accordingly.

Fig.3 The effects of JD5037 seem to be minimal, if any, in contrast to the conclusions of the authors. Densitometry should not be used for blots such as 3D. See our response to the general comments raised by this reviewer.

Figure 3J appears to have a transfer artifact in the Western, affecting the phospho-S6 lanes.

See our response to the general comments raised by this reviewer.

Figure 40 lacks controls for the cytoplasmic fraction. The levels of fibrillarlin differ dramatically - was this a control for the nuclear fraction? It is not mentioned in the text or legend. If this is a control, it brings into question the entire experiment since it is perhaps 100X lower in the control cells vs. the others. There is no clear difference between the relative localization of SREBP1 (cytoplasmic vs. nuclear) in any of the conditions. Was the densitometry done on multiple biologic replicates?

Thank you for this comment, which we decided to address by repeating our nuclear and cytoplasmic assessments of SREBP1 using Western Blot and Immunofluorescence analyses in hRPTCs exposed to high-glucose conditions in the presence/absence of JD5037 or rapamycin. As can be seen in the revised Figure 4s, t, the nuclear expression of SREBP1 is significantly increased in hyperglycemia, and it completely normalizes in JD5037- or rapamycin-treated cells. In this analysis fibrillarlin serves as a nuclear marker, and the blot represents four independent experiments. Additionally, quantification of cytoplasmic SREBP1 in the same conditions (see the new Figure 4u, v) was assessed by immunofluorescence staining, showing reduced cytoplasmic localization of SREBP1 in glucotoxic conditions, which was ameliorated in the drug-treated cells.

Figure 6 - if the mechanism is truly as proposed in Figure 7, the authors should treat the mice with Rapamycin or do a Raptor KO, and this should prevent CKD. The reviewer's statement basically ignores the published evidence, since we have recently shown that partial inhibition of mTORC1 in RPTCs, using specific RPTC-RPTOR KO mice, prevents fibrosis and a decline of renal function in diabetic mice [13]. Moreover, others have already shown that Rapamycin administration ameliorates renal injury in diabetic murine models [14-17]. Therefore, our current data strongly support the molecular mechanism proposed in our manuscript.

Reviewer #3 (Remarks to the Author):

This is a very elegant study in which the authors have described a typical example of how the endocannabinoid system, particularly when acting via CB₁ receptors, can play both physiological and pathological roles, as it is often the case with endogenous pro-homeostatic systems that can get deranged (and become maladaptive instead) during prolonged perturbations of homeostasis. The authors have elegantly dissected, in renal proximal tubular cells (RPTCs), almost all the molecular events

going from CB1 activation or overactivation and the final cellular response, which under physiological conditions, is concerned with regulation of amino acid uptake, and under conditions of hyperglycemia, with upregulation of glucose uptake. In the former case, the effect of CB1 is due to down-regulation of mTORC1 overactivity, whereas in the latter case to its overactivation.

We would like to deeply thank this Reviewer who found our study to be elegant and novel.

I have suggestions for only text corrections, nevertheless important to ameliorate the text:

1) Starting from the title, I had some occasional problems in following the syntax of the text. The title as such is not clear, and I suggest something like "Opposite physiological and pathological mTORC1-mediated roles of CB1 receptors in the control of renal tubular function". Likewise, sentences on page 4, lines 8-10, lines 13-14 and line 20, should be rephrased to afford clearer messages. On page 5, the long conclusive section of the Introduction is inappropriate for this part of the article, as it described the results of the study. It should be moved in the Results or Discussion, since the Introduction should only be used to provide the background and rationale for the study, and indicate the general experimental approach used to achieve the aims.

Thank you for this comment. The title was amended to the following: "Opposite Physiological and Pathological mTORC1-mediated Roles of the CB1 Receptor in Regulating Renal Tubular Function", and the introduction was changed and shortened according to the comment.

2) Not everybody knows what Akita mice are, and these guys are never mentioned in the Introduction. Therefore, I found the first mention of these mice on page 6, line 7, quite confusing.

Amended in the text (see page 5, line 10).

3) On page 7, the authors mention the results with measures of the levels of 2-AG biosynthetic precursor, 2-AG and 2-AG product, AA. I agree that the observed changes fit with the finding of elevated 2-AG biosynthesis, but also with its elevated degradation. This should be mentioned in the Discussion. If the authors also mentioned the expression of DAGL α and β in RPTCs, and found an increase of either or both enzymes following hyperglycemia, this would also fit nicely in the results. Indeed, Sampaio et al., 2015 (which should be quoted in this section of the Results too) did find that a model of RPTCs express more DAGL β than α , and it would be nice to see which of the two enzymes was responsible for 2-AG biosynthesis in hRPTCs;

Following this comment, we tested the gene and protein levels of both DAGL α and DAGL β in hRPTCs exposed to glucotoxic conditions, and except for a slight reduction in the DAGL α mRNA expression levels during hyperglycemia, no significant changes were found in the protein expression levels of the two isoforms (see the new data in Figure S5), further suggesting that acute hyperglycemia increases DAGL activity rather than expression. We added these findings to the Results section and discussed them as well as the Sampaio study in the discussion.

4) The Discussion is probably a bit too long, and yet the authors have not mentioned that it is not the first time that hyperglycemia leads to elevated levels of endocannabinoids in vitro, as shown for the first time by Matias et al. J Clin Endocrinol Metab 2006 in beta-cells. These authors also showed for the first time that diabetic hyperglycemic patients exhibit elevated AEA and 2-AG levels in the plasma, and therefore this paper and these previous findings should be mentioned;

The discussion was shortened, and we now refer to the work published by Matias et al., 2006.

5) On page 20, Line 16, the authors mention the possibility that CB1 may regulate Na⁺/K⁺-ATPase activity in RPTCs; indeed, this has been shown to be the case by Sampaio et al., *Biochem Pharmacol* 2018, and therefore also this paper should be mentioned.

The paper by Sampaio et al., 2018 is cited on page 15, line 1. In fact, we did not find any significant change in kidney Na⁺/K⁺ ATPase expression, either at the mRNA or protein levels in our RPTC-CB1 KO mice (Figure below a-c). Although a significant reduction in its mRNA expression was found in hRPTCs treated with the CB1R antagonist JD5037 for 24h, it did not result in changes in its protein levels (Figure below d-f). These findings may suggest that its activity may be regulated by CB1R, but additional work would need to further assess it.

6) As to the possible mechanism by which CB1 might lead to elevated 2-AG levels, this has been shown to occur via Gq-mediated elevation of intracellular calcium by Hegyi et al *Sci Rep* 2018, in astrocytes, whereas CB1 activation of the master regulator of 2-AG biosynthesis, phospholipase C beta, has been shown to occur again via Gq, in HEK-293 cells by Hermann et al, *Cell Mol Life Sci* 2003. These two papers should be quoted.

In fact, in our previous publication [9] we showed that CB1R activation in cultured primary hRPTCs induces intracellular calcium release (see Fig 3F in [7]), suggesting the possibility that CB1R may signal via Gq in RPTCs. However, in this manuscript we found that CB1R-induced activation of mTORC1 is most likely mediated via a G_{i/o} pathway. Interestingly, Mallipeddi and colleagues suggested that CB₁R activation can also stimulate phospholipase C via the G_{i/o} βγ subunit, leading to an increase in intracellular Ca²⁺ and activation of PKC [18]. All these papers including the Hermann et al. 2003 and Hegyi et al. 2018 are cited and discussed.

References:

1. Hanyaloglu, A. C. & Grammatopoulos, D. K. (2017) Pleiotropic GPCR signaling in health and disease, *Molecular and cellular endocrinology*. **449**, 1-2.
2. Afzal, M. S. (2020) G proteins: binary switches in health and disease, *Central-European journal of immunology*. **45**, 364-367.

3. Barutta, F., Corbelli, A., Mastrocola, R., Gambino, R., Di Marzo, V., Pinach, S., Rastaldi, M. P., Perin, P. C. & Gruden, G. (2010) Cannabinoid receptor 1 blockade ameliorates albuminuria in experimental diabetic nephropathy, *Diabetes*. **59**, 1046-54.
4. Jenkin, K. A., McAinch, A. J., Zhang, Y., Kelly, D. J. & Hryciw, D. H. (2015) Elevated cannabinoid receptor 1 and G protein-coupled receptor 55 expression in proximal tubule cells and whole kidney exposed to diabetic conditions, *Clin Exp Pharmacol Physiol*. **42**, 256-62.
5. Nam, D. H., Lee, M. H., Kim, J. E., Song, H. K., Kang, Y. S., Lee, J. E., Kim, H. W., Cha, J. J., Hyun, Y. Y., Kim, S. H., Han, S. Y., Han, K. H., Han, J. Y. & Cha, D. R. (2012) Blockade of cannabinoid receptor 1 improves insulin resistance, lipid metabolism, and diabetic nephropathy in db/db mice, *Endocrinology*. **153**, 1387-96.
6. Jourdan, T., Szanda, G., Rosenberg, A. Z., Tam, J., Earley, B. J., Godlewski, G., Cinar, R., Liu, Z., Liu, J., Ju, C., Pacher, P. & Kunos, G. (2014) Overactive cannabinoid 1 receptor in podocytes drives type 2 diabetic nephropathy, *Proc Natl Acad Sci U S A*. **111**, E5420-8.
7. Udi, S., Hinden, L., Earley, B., Drori, A., Reuveni, N., Hadar, R., Cinar, R., Nemirovski, A. & Tam, J. (2017) Proximal Tubular Cannabinoid-1 Receptor Regulates Obesity-Induced CKD, *J Am Soc Nephrol*. **28**, 3518-3532.
8. Jourdan, T., Park, J. K., Varga, Z. V., Paloczi, J., Coffey, N. J., Rosenberg, A. Z., Godlewski, G., Cinar, R., Mackie, K., Pacher, P. & Kunos, G. (2018) Cannabinoid-1 receptor deletion in podocytes mitigates both glomerular and tubular dysfunction in a mouse model of diabetic nephropathy, *Diabetes Obes Metab*. **20**, 698-708.
9. Hinden, L., Udi, S., Drori, A., Gammal, A., Nemirovski, A., Hadar, R., Baraghithy, S., Permyakova, A., Geron, M., Cohen, M., Tsytkin-Kirschenschweig, S., Riahi, Y., Leibowitz, G., Nahmias, Y., Priel, A. & Tam, J. (2018) Modulation of Renal GLUT2 by the Cannabinoid-1 Receptor: Implications for the Treatment of Diabetic Nephropathy, *J Am Soc Nephrol*. **29**, 434-448.
10. Marks, J., Carvou, N. J., Debnam, E. S., Srai, S. K. & Unwin, R. J. (2003) Diabetes increases facilitative glucose uptake and GLUT2 expression at the rat proximal tubule brush border membrane, *The Journal of physiology*. **553**, 137-45.
11. Cohen, M., Kitsberg, D., Tsytkin, S., Shulman, M., Aroeti, B. & Nahmias, Y. (2014) Live imaging of GLUT2 glucose-dependent trafficking and its inhibition in polarized epithelial cysts, *Open Biol*. **4**.
12. Vallon, V., Rose, M., Gerasimova, M., Satriano, J., Platt, K. A., Koepsell, H., Cunard, R., Sharma, K., Thomson, S. C. & Rieg, T. (2013) Knockout of Na-glucose transporter SGLT2 attenuates hyperglycemia and glomerular hyperfiltration but not kidney growth or injury in diabetes mellitus, *American journal of physiology Renal physiology*. **304**, F156-67.
13. Kogot-Levin, A., Hinden, L., Riahi, Y., Israeli, T., Tirosh, B., Cerasi, E., Mizrachi, E. B., Tam, J., Mosenzon, O. & Leibowitz, G. (2020) Proximal Tubule mTORC1 Is a Central Player in the Pathophysiology of Diabetic Nephropathy and Its Correction by SGLT2 Inhibitors, *Cell Rep*. **32**, 107954.
14. Xiao, T., Guan, X., Nie, L., Wang, S., Sun, L., He, T., Huang, Y., Zhang, J., Yang, K., Wang, J. & Zhao, J. (2014) Rapamycin promotes podocyte autophagy and ameliorates renal injury in diabetic mice, *Mol Cell Biochem*. **394**, 145-54.
15. Yang, Y., Wang, J., Qin, L., Shou, Z., Zhao, J., Wang, H., Chen, Y. & Chen, J. (2007) Rapamycin prevents early steps of the development of diabetic nephropathy in rats, *American journal of nephrology*. **27**, 495-502.
16. Yu, R., Bo, H., Villani, V., Spencer, P. J. & Fu, P. (2016) The Inhibitory Effect of Rapamycin on Toll Like Receptor 4 and Interleukin 17 in the Early Stage of Rat Diabetic Nephropathy, *Kidney & blood pressure research*. **41**, 55-69.

17. Mori, H., Inoki, K., Masutani, K., Wakabayashi, Y., Komai, K., Nakagawa, R., Guan, K. L. & Yoshimura, A. (2009) The mTOR pathway is highly activated in diabetic nephropathy and rapamycin has a strong therapeutic potential, *Biochem Biophys Res Commun.* **384**, 471-5.
18. Mallipeddi, S., Janero, D. R., Zvonok, N. & Makriyannis, A. (2017) Functional selectivity at G-protein coupled receptors: Advancing cannabinoid receptors as drug targets, *Biochem Pharmacol.* **128**, 1-11.

REVIEWER COMMENTS

Reviewer #1 (Remarks to the Author):

The authors clarify the role of CB1R-mTORC1 axis in proximal tubular cells in regulating renal function in diabetic and non-diabetic condition.

Reviewer #2 (Remarks to the Author):

Hinden et al.

There continue to be concerns about data quality, and the authors have not attempted to provide clearer blots. As an example, Figure 2N "Immunoblotting analysis and quantification of GLUT2 in hRPTCs treated with HG (30 22 mM) or HG + JD5037 (100nM) for 24 h. n = 5-6 biological replicates per group" shows widely variable levels of Glut2, especially in the controls but also in the treated samples. This is also true in 2U and elsewhere.

The authors state that the many gel artifacts do not overlap with the bands, so they do not raise major concerns, but this is not the case for example in Figure 3D, where the artifacts obscure the ph-Akt band yet densitometric analyses were still performed, and Figure 3J where the phospho-S6 bands appear to be obscured (compare 6th band from right to 7th band from right). The gel in Figure 4S appears to have transferred poorly.

While some variation is expected for in vivo specimens, this level of variability is unusual for in vitro experimentation. These concerns limit the interpretation of some aspects this work.

Reviewer #3 (Remarks to the Author):

The authors have addressed all my comments and requests for clarifications and data in a very satisfactory manner.

Reviewer #1 (Remarks to the Author):

The authors clarify the role of CB1R-mTORC1 axis in proximal tubular cells in regulating renal function in diabetic and non-diabetic condition.

We would like to thank this reviewer for his/her constructive comments that raised the quality of our work.

Reviewer #2 (Remarks to the Author):

Hinden et al. There continue to be concerns about data quality, and the authors have not attempted to provide clearer blots.

We now provide improved western blots for all the experiments highlighted by this reviewer.

As an example, Figure 2N “Immunoblotting analysis and quantification of GLUT2 in hRPTCs treated with HG (30 22 mM) or HG + JD5037 (100nM) for 24 h. n = 5-6 biological replicates per group” shows widely variable levels of GLUT2, especially in the controls but also in the treated samples. This is also true in 2U and elsewhere.

We repeated the experiment described in Figure 2N three additional times, and its quantification, reported in Figure 2P, describes now 8-9 independent biological replicates per group. The expression of GLUT2 in hRPTCs was also tested via immunohistochemistry (Figure 2O and Q) and similar expression profile of GLUT2 was found, and thus our findings are valid.

In addition, we repeated the experiment described in Figure 2U, and its quantification, presented in Figure 2V, describes now 8 independent biological replicates per group.

The authors state that the many gel artifacts do not overlap with the bands, so they do not raise major concerns, but this is not the case for example in Figure 3D, where the artifacts obscure the p-Akt band yet densitometric analyses were still performed.

We repeated the experiment described in Figure 3D three additional times, and its quantification, reported in Figure 3E and F, describes now 12-16 independent biological replicates per group.

and Figure 3J where the phospho-S6 bands appear to be obscured (compare 6th band from right to 7th band from right).

We repeated the experiment described in Figure 3J three additional times, and its quantification, reported in Figure 3K and L, describes now 8-9 independent biological replicates per group.

The gel in Figure 4S appears to have transferred poorly.

We rerun the western blot describing the experiment in Figure 4S. The current results show a good transfer of all the samples.

While some variation is expected for *in vivo* specimens, this level of variability is unusual for *in vitro* experimentation. These concerns limit the interpretation of some aspects this work.

Please note that all the *in vitro* experiments were done in primary human RPTCs, and therefore some variations are also expected in such experiments. Nevertheless, we hope that by providing the improved blots, which represent additional biological repetitions of these experiments, the reviewer will appreciate the novelty of our findings and their interpretation.

Reviewer #3 (Remarks to the Author):

The authors have addressed all my comments and requests for clarifications and data in a very satisfactory manner.

We would like to thank this reviewer for his/her valuable comments that improved the quality of our work.

REVIEWERS' COMMENTS

Reviewer #2 (Remarks to the Author):

no additional concerns, the new blots and analyses are an improvement

Response to the Reviewers: Ms. Ref. #NCOMMS-20-44454B

Reviewer #2 (Remarks to the Author):

No additional concerns, the new blots and analyses are an improvement

We would like to thank the reviewer for finding our blots and analysis an improvements.